# UNPACKING HUMAN PREFERENCE FOR LLMS: DEMOGRAPHICALLY AWARE EVALUATION WITH THE HUMAINE FRAMEWORK

**Nora Petrova**[*]    **Andrew Gordon**[*]    **Enzo Blindow**

Prolific

## ABSTRACT

The evaluation of large language models faces significant challenges. Technical benchmarks often lack real-world relevance, while existing human preference evaluations suffer from unrepresentative sampling, superficial assessment depth, and single-metric reductionism. To address these issues, we introduce HUMAINE, a framework for multidimensional, demographically aware measurement of human-AI interaction. We collected multi-turn, naturalistic conversations from 23,404 participants that were stratified across 22 demographic groups, both in the US and UK, to evaluate 28 state-of-the-art models across five human-centric dimensions. We use a hierarchical Bayesian Bradley-Terry-Davidson (BTD) model, with post-stratification to census data, and our analysis reveals three key insights. **(1)** We establish a clear performance hierarchy where `google/gemini-2.5-pro` ranks first overall, with a 95.6% posterior probability of being the top-ranked model. **(2)** We uncover significant preference heterogeneity, with user age emerging as the primary demographic axis of disagreement; a model's perceived rank can shift substantially across age groups, exposing failures in generalisation that unrepresentative samples typically mask. **(3)** We quantify the vast difference in discriminative power across evaluation dimensions, with ambiguous qualities like *Trust, Ethics & Safety* showing a 65% tie rate, in stark contrast to the decisive 10% tie rate for *Overall Winner*. Our work emphasises the need for a more multidimensional, demographically aware perspective in LLM evaluation. We release our complete dataset, interactive leaderboard, and open-source framework.

🤗 **Leaderboard:** huggingface.co/spaces/ProlificAI/humaine-leaderboard
🤗 **Dataset:** huggingface.co/datasets/ProlificAI/humaine-evaluation-dataset

## 1 INTRODUCTION

Large Language Models (LLMs) have facilitated a sea of change in how humans interact with AI, and have become deeply integrated into professional workflows, personal decisions, and creative tasks. However, this rapid progress has created a critical "evaluation gap", where our methods for measuring models have not kept pace with their real-world impact. This gap is perpetuated by the pervasiveness of automated benchmarks, which almost exclusively assess technical performance while overlooking how the systems resonate with the people who actually use them (Bowman & Dahl, 2021). As a result, optimising for benchmarks alone risks developing models that are technically impressive but fail to meet human needs and expectations (Amershi et al., 2019), and leave the entire AI ecosystem without the reliable human-centric data needed to guide responsible AI development and deployment.

The field's dependence on automated benchmarks exemplifies this problem. Benchmarks like MMLU (Hendrycks et al., 2021), HELM (Liang et al., 2022), and BIG-Bench (Srivastava et al., 2022) are indispensable for establishing a model's technical floor by assessing its foundational reasoning and knowledge. However, their design as standardised tests makes them blind to the subjective, dynamic qualities of conversation. They fall short of measuring a model's ability to maintain

---

[*]Correspondence to: {nora.petrova, andrew.gordon}@prolific.com

context, adapt its tone, or build user trust. As the field orients itself around these metrics, development can fall prey to a form of "Goodhart's Law", where optimising for the benchmark becomes the goal, rather than improving the holistic user experience the score was intended to represent. Ultimately, while these benchmarks measure what a model knows, they fail to capture how it behaves in the complex domain of human collaboration.

To address the limits of automated benchmarks, a second paradigm has emerged of direct human preference evaluation. Influential platforms like Chatbot Arena (Zheng et al., 2023) represent a crucial step forward by crowdsourcing pairwise comparisons from users in live conversations. However, their approach is undermined by foundational methodological flaws. First, their reliance on a self-selected, anonymous user base leads to unrepresentative sampling. Second, judgments often based on minimal interaction result in superficial assessment depth. Finally, binary preference votes create single-metric reductionism, obscuring the multidimensional nature of interaction quality. These inherent issues are compounded by systemic artefacts; as documented by Singh et al. (2025), practices like undisclosed private testing and evaluation gaming can distort rankings independently of true model quality.

To resolve this double bind, where benchmarks miss human nuance and existing preference evaluations lack scientific rigour, we introduce HUMAINE: a framework designed for multidimensional, demographically aware measurement of human-AI interaction. HUMAINE's methodology is based on the foundational principles of psychometric measurement and directly addresses the flaws of existing approaches.

Our primary contributions are:

1. **The HUMAINE Framework:** A methodology for human-centric AI evaluation that addresses three validity threats in existing approaches: sampling bias, assessment depth, and metric reductionism.

2. **A Large-Scale, Demographically Stratified Dataset:** 119,890 multi-dimensional human judgments from 23,404 participants across 28 models, plus structured metadata characterising conversational dynamics, task properties, and interaction outcomes.

3. **Key Empirical Insights:** Our analysis reveals how model rankings shift across demographic groups and evaluation dimensions, with implications for context-appropriate model selection.

4. **A Living Evaluation Framework:** A regularly updated leaderboard tracking state-of-the-art model performance as new models are released.

## 2 RELATED WORKS

The HUMAINE framework is situated at the intersection of several research domains: LLM evaluation, psychometric measurement theory, human-computer interaction (HCI), and the study of fairness and representation in AI.

### 2.1 PARADIGMS IN LLM EVALUATION

Current LLM evaluation is primarily characterised by two paradigms. The first, automated benchmarks, provides essential measures of technical capability through standardised tests like MMLU (Hendrycks et al., 2021), HELM (Liang et al., 2022), AGIEval (Zhong et al., 2023), and BIG-Bench (Srivastava et al., 2022). HUMAINE complements this work by addressing their inability to capture subjective interaction quality (Bowman & Dahl, 2021). The second, human preference evaluation, pioneered by RLHF approaches (Ouyang et al., 2022) and platforms like Chatbot Arena (Zheng et al., 2023) and the Open LLM Leaderboard (Fourrier et al., 2023), moved evaluation towards real-world interaction. Our framework is a direct response to the significant methodological flaws of this approach, including unrepresentative sampling and systemic gaming (Singh et al., 2025).

A third emerging paradigm is model-based evaluation, or "LLM-as-a-judge" (Zheng et al., 2023; Liu et al., 2023). While this approach offers scalability, it is prone to a host of biases. HUMAINE adopts a complementary role for the LLM judge instead: not as a proxy for human preference, but as

a tool for structured, post-hoc analysis of conversations to help explain the mechanisms underlying human judgments.

## 2.2 PSYCHOMETRIC FOUNDATIONS FOR PREFERENCE MODELLING

The conversion of pairwise comparisons into a continuous scale has a long history in psychometrics, originating with Thurstone's Law of Comparative Judgment (Thurstone, 1927) and formalised in models like the Bradley-Terry model (Bradley & Terry, 1952). The statistical engine of HUMAINE, a hierarchical Bayesian implementation of the Bradley-Terry-Davidson model, applies modern statistical techniques to this established measurement framework, allowing for robust uncertainty quantification and the modelling of complex, multi-level effects.

## 2.3 HUMAN-CENTRIC AND USABILITY FRAMEWORKS

HUMAINE's multi-dimensional metrics are grounded in decades of research from Human-Computer Interaction (HCI). The framework aligns with concepts like the Technology Acceptance Model (TAM), which posits that "perceived usefulness" and "perceived ease of use" are primary determinants of technology adoption (Davis, 1989). Our dimensions map directly onto these ideas: "Core Task Performance" reflects usefulness, while "Interaction Fluidity" and "Communication Style" capture ease of use. This approach operationalises principles from the *Guidelines for Human-AI Interaction* (Amershi et al., 2019), which emphasise that AI systems must be understandable, adaptable, and trustworthy.

## 2.4 REPRESENTATIVE DATA AND FAIRNESS IN AI EVALUATION

A core contribution of HUMAINE is its commitment to representative sampling. The reliance on unrepresentative datasets has been shown to result in systems that perform inequitably across demographic groups, as famously demonstrated in facial recognition by Buolamwini & Gebru (2018). More recently, Santurkar et al. (2023) provided direct empirical evidence that rater demographics significantly impact preferences for LLM behaviour, showing that an aggregate score can mask important disagreements between populations. This aligns with findings from Kirk et al. (2024), who demonstrated the importance of demographically diverse preference data for safety alignment.

Recent work has further highlighted the complexities of human preference data for AI evaluation. Li et al. (2024) examine the relationship between human and LLM preferences, revealing systematic differences in how different evaluators assess model outputs, where humans prioritise stance-alignment and are less sensitive to errors, while advanced LLMs emphasise correctness and harmlessness. Hosking et al. (2024) demonstrate that human feedback systematically underrepresents crucial error types like factuality and is biased by output assertiveness, challenging the assumption that preference scores serve as a reliable singular gold standard. These findings underscore the importance of understanding preference heterogeneity and employing multidimensional evaluation rather than treating human judgments as monolithic.

HUMAINE addresses these sampling challenges through demographically stratified recruitment with post-stratification adjustments to census data. While this initial implementation focuses on US and UK populations, the framework is designed to be extensible to additional geographies, languages and demographic dimensions.

## 3 METHODOLOGY

### 3.1 MODEL & PARTICIPANT SELECTION

We selected 28 state-of-the-art language models representing the current frontier of conversational AI, accessed via `openrouter.ai` with default settings. As HUMAINE is designed as a living benchmark, we add new models and update rankings as they become available; the list of models is therefore a snapshot at the time of writing.

We recruited 23,404 participants through the Prolific platform, compensating them at the recommended rate of £9/hr. To enable deep demographic analysis, we stratified our sampling to include

22 specific demographic strata across geographic location (UK/US), age (18-34, 35-54, 55+), ethnicity (Asian, Black, White and Other in the UK, and Hispanic, Asian, African American, and White in the US), and political affiliation (Democrat, Republican, and Independent in the US, and Conservative, Labour, Liberal Democrats, Greens, and Reform UK in the UK). For more detail on the participant experience, refer to Appendix B.

## 3.2 Data Collection and Benchmark Design

The HUMAINE benchmark employs a pairwise comparison framework. Participants were presented with two anonymised models side-by-side and were free to select their own conversation topic. To ensure sufficient interaction depth, a minimum of 3 conversational turns was required. Each message sent by the participant was delivered to both models simultaneously, ensuring identical user-side conversation flow for a controlled comparison. This design choice ensures participants compare models on their handling of identical conversational contexts rather than on different conversation trajectories. Independent conversations risk diverging in ways that make fair pairwise comparison impossible, for instance, if Model A gives a brief factual answer while Model B provides an expansive tutorial, the participant's follow-up questions would naturally differ, creating distinct conversational trajectories that confound preference judgments.

We recruited participants across 22 demographic strata with balanced sample sizes, resulting in 1,848 to 2,636 comparisons per stratum with a median conversation length of 6 turns. Detailed sample sizes for all strata can be calculated from the released feedback dataset.

To maximise data collection efficiency, model pairings were determined by a TrueSkill-based adaptive sampling algorithm (Herbrich et al., 2006). By maintaining skill and uncertainty estimates for each model, the algorithm strategically selects matchups where the outcome is most uncertain, thereby maximising information gain and accelerating the convergence of rankings.

Each of our 22 demographic strata was run as a dedicated TrueSkill tournament. Participants could qualify for and participate in multiple tournaments corresponding to their demographic profile (e.g., a Hispanic, 18-34 Democrat could participate in three separate tournaments). Additionally, participants could take part in multiple data collection batches, receiving new, randomly assigned model pairs each time. The statistical implications of this multi-membership design are handled by our hierarchical model, as detailed in Subsection 3.4.1.

Conversation quality was monitored in real-time by a `gpt-4o-mini` judge that flagged low-effort inputs (e.g., single-word responses, repetitive copy-pasted messages, or clear disengagement), providing constructive feedback to participants. Participants receiving three warnings were removed from the study. This affected less than 1.6% of our sample. After the conversation, participants evaluated the two models across the five comparative dimensions, selecting their preferred model or indicating a tie.

More details on data collection and the quality assurance mechanism are available in Appendix B.

## 3.3 Evaluation Metrics

Our four evaluation dimensions were derived from a pilot study using factor analysis to identify the core drivers of user preference. To these four dimensions, we added a holistic overall winner metric:

- **Core Task Performance & Reasoning:** How effectively the model accomplishes tasks and demonstrates sound reasoning and understanding.
- **Communication Style & Presentation:** The model's language, tone, personality, and the appropriateness of its detail and intuitiveness.
- **Interaction Fluidity & Adaptiveness:** How smoothly and adaptively the model interacts, manages conversation flow, and responds to user input.
- **Trust, Ethics & Safety:** The reliability, transparency, ethical conduct, and safety of the model's outputs and behaviour.
- **Overall Winner:** A holistic preference judgement incorporating all aspects of the interaction.

For more information on the pilot study, please refer to Appendix E.

### 3.4 ANALYSIS FRAMEWORK

#### 3.4.1 HIERARCHICAL BRADLEY-TERRY-DAVIDSON MODEL

We develop a hierarchical Bayesian Bradley-Terry-Davidson (BTD) model to convert pairwise comparisons into continuous skill ratings. The model extends the classic BT framework to handle ties and captures demographic heterogeneity through a factorised structure. At its core, it learns a global skill parameter ($\theta$) for each model-metric combination, then adds demographic-specific adjustments ($u$). These adjustments are hierarchically modelled with heterogeneity parameters ($\tau$) that quantify the magnitude of preference variation. The model outputs posterior distributions for all parameters, enabling us to derive overall leaderboards, demographic-specific rankings, and measures of preference heterogeneity.

**Disentangling Mixed Demographic Effects with Hierarchical Modelling.** A key challenge of our participant design is that a single participant's preference (e.g., from someone who is Asian, 18-34, and a Democrat) could be driven by any of their demographic identities. Our single, unified hierarchical model is designed to disentangle these mixed effects. The tournament structure is solely a device during data-collection, and for the analysis all comparisons are pooled.

The model represents each participant by their position on three demographic axes (age, ethnicity, politics). The model then learns two components simultaneously through partial pooling: a global skill parameter for each model, and a set of additive adjustments for each demographic group. These group adjustments are centred within each axis, allowing them to be interpreted as deviations from the average preference for that axis. Critically, this design allows the model to attribute a consistent preference pattern to the correct demographic driver, even when a participant belongs to several groups, while leveraging the entire dataset to ensure the global skill estimates remain robust.

Full mathematical details are provided in Appendix A.

#### 3.4.2 LLM JUDGE FOR CONVERSATIONAL ANALYSIS

To provide a deeper, quantitative understanding of the conversations underlying human preferences, we conducted a post-hoc analysis of all conversation transcripts using an LLM judge.

**Model Selection and Justification.** For this role, we selected `gpt-4.1`, balancing three key considerations: performance, practicality, and precedent. The model offered a strong trade-off between state-of-the-art instruction-following and the inference speed needed to process our large dataset. Its availability via a stable API was critical for reproducibility, and its capabilities in similar annotation tasks have been demonstrated in prior literature and validated by our internal testing.

**Procedural Safeguards.** While acknowledging that no LLM-based analysis can be entirely free of bias, we implemented several procedural safeguards to ensure the integrity and utility of the outputs. The core principle of our approach was strict separation where the LLM analysis was conducted entirely post-hoc, was never used to generate competitive rankings, and had no influence on the primary human preference scores. Its role was purely explanatory. To enhance reliability, we used a detailed, structured prompt with explicit rubrics. This approach allows us to leverage the scalability of LLMs to generate rich metadata that characterises conversational dynamics, task properties, and outcomes as an explanatory tool for understanding human judgments. Full details on the metrics, categories, and prompt are provided in Appendix C.

## 4 RESULTS

We structure our findings into four parts: (1) we establish the overall model performance leaderboard; (2) we quantify the significant heterogeneity in preferences across demographic groups; (3) we examine how model rankings vary by evaluation dimension; and (4) we assess the discriminative power of each metric.

### 4.1 OVERALL MODEL PERFORMANCE

Our primary result is a robust ranking of the 28 models derived from our hierarchical BTD model, post-stratified to US and UK census data. The main performance metric, shown in Figure 1, is

the Score (Winshare). This score represents a model's expected total points from a round-robin tournament against all other models, where a win is worth 1 point and a tie is worth 0.5, for a maximum possible score of 27.

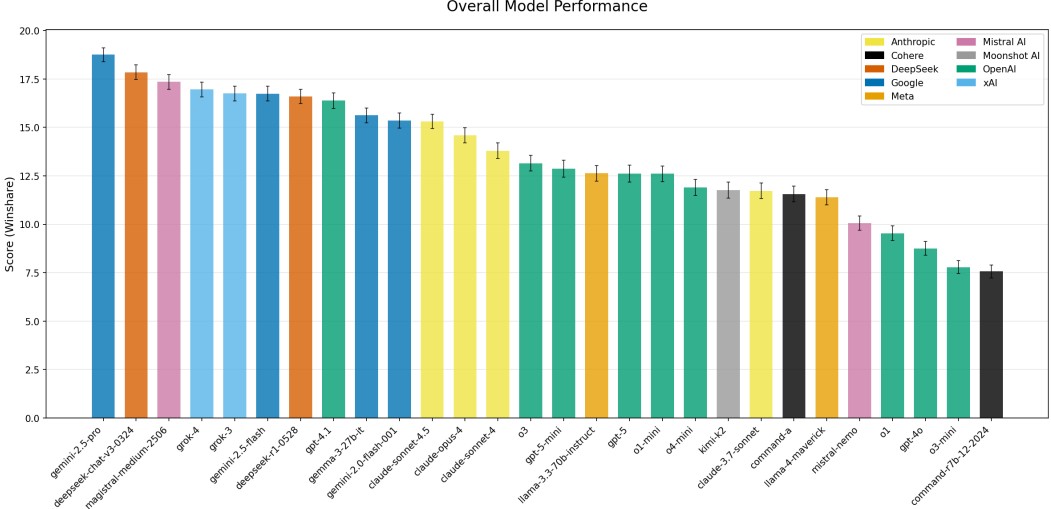

Figure 1: Model performance on the "Overall Winner" metric. Bars represent the Score (expected points in a round-robin tournament; max=27, mean=13.5), with 95% credible intervals.

`google/gemini-2.5-pro` stands out as the clear winner. Its leading position is substantiated not only by its top score, but also by the high statistical confidence assigned to its rank. Our Bayesian model calculates a 95.6% probability of it being the best model (P(best)). A distinct gap separates it from the next competitor, `deepseek/deepseek-chat-v3-0324`, which in turn holds a lead over the subsequent tier of models.

Below the top two, a competitive group including `mistralai/magistral-medium-2506`, `x-ai/grok-4`, and `x-ai/grok-3` emerges with closely overlapping credible intervals. This establishes a clear hierarchy at the top of the leaderboard that progressively flattens, rendering many lower-ranked models statistically indistinguishable.

## 4.2    DEMOGRAPHIC HETEROGENEITY IN MODEL PREFERENCE

A key objective of the HUMAINE framework is to move beyond a single aggregate leaderboard to quantify how model preferences vary across different populations. Our analysis reveals that, among the three demographic axes we studied, age is the most significant factor driving preference heterogeneity, substantially exceeding the effects of ethnicity and political affiliation. While our hierarchical model quantifies this through a latent heterogeneity parameter ($\tau$), the practical impact of this finding is best understood through the two more interpretable metrics visualised in Figure 2.

The **left panel** shows that a model's average rank shifts (defined as the mean absolute difference between a model's rank in a given demographic group and its overall rank across all groups) by a substantial ±2.8 ranks across age cohorts, a far larger variance than for ethnicity (±1.3) or political affiliation (±1.5). To illustrate this divergence: `mistralai/magistral-medium-2506` is a clear favourite among younger users (ranking 1st in the US and 2nd in the UK for the 18-34 cohort) but drops precipitously with older users (falling to 5th in the US and 10th in the UK for the 55+ cohort). Conversely, `google/gemini-2.5-pro` sees its standing improve with age, securing the top rank across older cohorts in both regions. The **right panel** reveals a clear trend in user decisiveness: tie rates (the proportion of pairwise comparisons where participants selected "Tie" rather than declaring a winner) increase steadily with age, from 9.7% for the 18-34 cohort to 12.5% for users aged 55+, representing a 29% rise in indecisiveness. Together, these findings empirically validate our central claim that a single aggregate leaderboard is insufficient, as it masks meaningful variations in both inter-group agreement and user decisiveness.

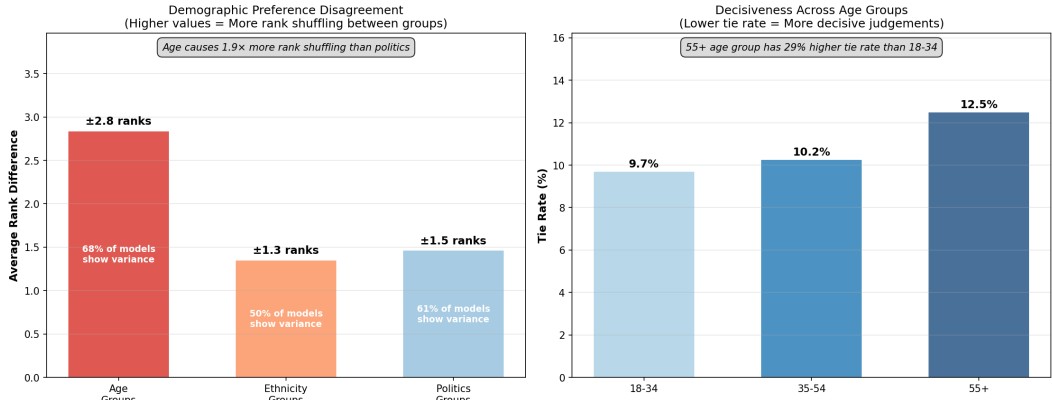

Figure 2: Demographic preference heterogeneity, shown by: **(Left)** inter-group disagreement (avg. rank difference), and **(Right)** user decisiveness (tie rates by age).

## 4.3 PERFORMANCE ACROSS EVALUATION DIMENSIONS

While the overall leaderboard provides a valuable summary, it obscures important nuances in model performance. The rankings across all five evaluation dimensions reveal that a model's competitive standing can change meaningfully depending on the evaluation lens.

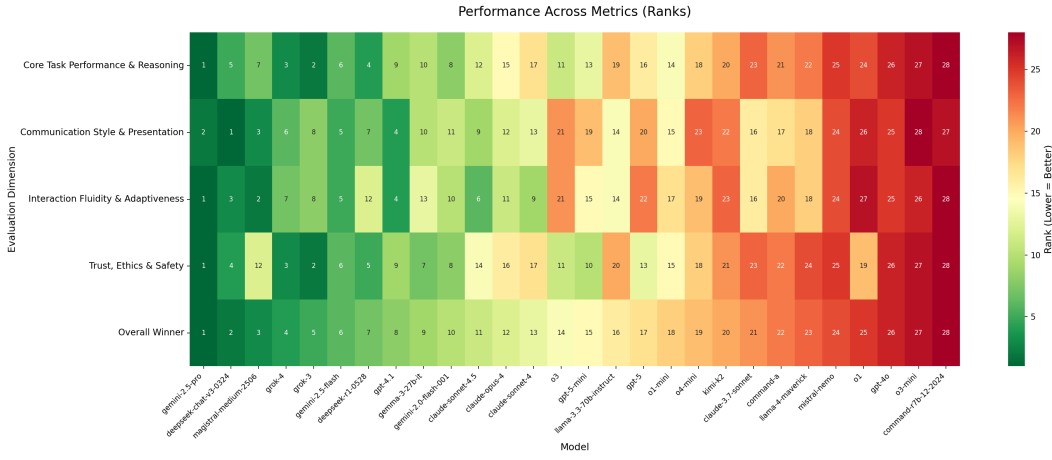

Figure 3: Heatmap showing model rankings across five evaluation dimensions. Lower ranks (darker green) indicate better performance. Models show significant variation in their relative strengths, with some excelling in reasoning while others lead in communication or trust.

While `google/gemini-2.5-pro` maintains the top position across all dimensions, the rankings of other models shift significantly. Notably, `x-ai/grok-3` performs substantially better on *Core Task Performance & Reasoning* (ranking 2nd) than on *Communication Style & Presentation* or *Interaction Fluidity & Adaptiveness* (ranking 8th on both). Conversely, `mistralai/magistral-medium-2506` excels in *Interaction Fluidity & Adaptiveness* (ranking 2nd) but ranks lower in *Core Task Performance & Reasoning* and *Trust, Ethics & Safety* (ranking 7th and 12th respectively). These shifts underscore the multi-faceted nature of human preference and highlight the risk of relying on a single "overall" score for model selection.

## 4.4 DISCRIMINATIVE POWER OF EVALUATION DIMENSIONS

The BTD model's tie-propensity parameter ($\nu_k$) allows us to assess the discriminative power of each evaluation dimension. We observe substantial variation in how decisively participants can distinguish between models across different metrics.

Figure 4: Discriminative power of evaluation dimensions measured by tie rates. Trust, Ethics & Safety shows the highest ambiguity (65% ties), while Overall Winner is most decisive (10% ties).

*Trust, Ethics & Safety* was the least discriminative dimension with a 65% tie rate, suggesting either model convergence on safety or that the quality is inherently difficult to assess within the current interaction setup. In stark contrast, *Overall Winner* was the most discriminative (10% ties), indicating that users can form decisive holistic preferences even when specific attributes are ambiguous.

This metric hierarchy suggests that the effectiveness of pairwise comparisons is highly dependent on the evaluation metric. Holistic judgments like *Overall Winner* provide a strong, decisive signal in open-ended conversations. Conversely, the high ambiguity of *Trust, Ethics & Safety* indicates a more tailored approach is needed to elicit relevant behaviours.

### 4.5 CONVERSATIONAL ANALYSIS

We conducted post-hoc LLM-based classification to characterise the dataset across task types, domains, and interaction quality metrics (task complexity, goal achievement, user engagement). This analysis provides transparency into what types of conversations participants had and demonstrates the breadth of interaction contexts captured by the open-ended design.

**Task Types.** The majority of conversations involved information seeking (71.5%), followed by personal advice (10.5%), with smaller proportions of project planning (2.7%), technical assistance (2.4%), and decision support (2.2%).

**Domains.** Conversations spanned 41 distinct domains. The most common were health/medical (12.9%), sports (8.8%), technology (8.1%), cooking/food (7.6%), and creative arts (7.5%).

**Task Complexity.** Rated on a 5-point scale (mean 3.54, median 4.0), with 43.2% of conversations rated as moderately complex (score 4) and 12.3% as highly complex (score 5).

**Goal Achievement.** Also rated on a 5-point scale (mean 4.32, median 4.0), heavily right-skewed with 92.6% of conversations rated as achieving their intended purpose (scores 4-5).

**User Engagement.** Rated on a 5-point scale (mean 3.30, median 3.0), with 70.2% showing moderate engagement (score 3) and 24.6% showing high engagement (score 4).

These patterns confirm that the evaluation captured real-world conversational interactions spanning diverse topics, task types, and complexity levels.

## 5 DISCUSSION

The HUMAINE framework advances the evaluation of LLMs by moving beyond single-metric leaderboards to reveal the multidimensional and demographically contingent nature of human-AI interaction quality. Our analysis uncovered three key insights that challenge current evaluation paradigms and offer a new path forward for model development, assessment, and selection.

First, **dimensional analysis reveals that "best" is a context-dependent illusion.** While confirming that `google/gemini-2.5-pro` leads across multiple dimensions, HUMAINE demonstrates *why* it succeeds: through consistency and balanced high performance. Our findings show

models exhibit different competitive standings depending on the evaluation lens. For example, `deepseek-chat-v3-0324` leads on *Communication Style & Presentation* but ranks 5th on *Core Task Performance & Reasoning*, while `mistralai/magistral-medium-2506` excels in *Interaction Fluidity & Adaptiveness* (ranking 2nd) but lags in *Core Task Performance & Reasoning* and *Trust, Ethics & Safety* (ranking 7th and 12th respectively). This multifaceted performance becomes particularly striking when contrasted with technical benchmarks like HELM, where `google/gemini-2.5-pro` currently ranks a modest 13th, a dramatic disparity highlighting the evaluation gap between technical accuracy and human preference. These insights disappear when collapsed into a single number, leaving users unable to discern if a model's success is due to its reasoning power, communication skills, or balanced competence. HUMAINE moves the conversation from *"which model is best?"* to *"best for what and for whom?"* and demonstrates that meaningful model selection requires aligning specific dimensional strengths with intended use cases.

Second, **the discovery that age is the primary driver of preference heterogeneity exposes a demographic blind spot in AI development.** Current evaluation practices that rely on unrepresentative user bases systematically obscure these important performance gaps. Our analysis reveals that user preferences shift by ±2.8 ranks across age cohorts, far exceeding variation for ethnicity (±1.3) or politics (±1.5). Strikingly, the pattern of tie rates reveals how age shapes evaluation certainty, while all age groups show similar decisiveness about *Communication Style & Presentation* (17-20% ties), older users become progressively less certain about *Core Task Performance & Reasoning* (rising from 32% ties for 18-34 to 39% for 55+). This suggests younger users have clearer expectations for functional capability, while older users find it harder to distinguish between models on core utility, potentially reflecting different mental models of what AI should accomplish. The overall trend toward higher tie rates among older users (from 9.7% for 18-34 to 12.5% for 55+) indicates that qualities differentiating models for younger demographics are systematically less salient for other groups. These findings suggest that models tuned on narrow, tech-savvy feedback risk creating preference optimisation loops that systematically exclude broader populations, undermining both market adoption and equitable performance.

Third, **the vast difference in metric discriminability reveals that evaluation methodologies should be tailored to the constructs they aim to measure.** Our analysis shows that the context of an interaction is critical for reliably assessing certain qualities. The 65% tie rate for *Trust, Ethics & Safety* suggests that these qualities are not consistently elicited in open-ended, user-driven conversations, making them difficult for participants to meaningfully compare. This finding offers a clear methodological principle for the field, while broad-based preference testing like HUMAINE is highly effective for measuring general utility, assessing critical but nuanced attributes like perceived safety demands a move towards more specialised interaction scenarios that create the necessary context for meaningful judgment.

## 5.1 LIMITATIONS AND FUTURE WORK

While our methodology addresses key flaws in existing paradigms, we acknowledge several limitations. The current instantiation focuses on US and UK populations due to budget constraints and census data availability for post-stratification. This limits global applicability as cultural context can profoundly influence preferences. Our demographic stratification focused on three axes (age, ethnicity, political affiliation) and could be extended to include factors like gender, education, and socioeconomic status, which may reveal additional layers of preference heterogeneity. We leave it to future work to expand to non-Western cultures and include these additional demographic factors.

Our focus on short, multi-turn conversations cannot capture long-term phenomena like persona consistency or performance degradation over extended dialogues. Moreover, the open-ended nature of the tasks means that task complexity was not controlled.

The five evaluation dimensions, while empirically derived, may not be exhaustive. Qualities like creativity, humour, or empathy may be significant preference drivers in certain contexts, and the importance of these dimensions may vary across cultures.

The HUMAINE framework is currently limited to text-only interactions. This represents a growing gap, as the state-of-the-art models under evaluation are increasingly capable of processing and generating images, audio, and other data types. A text-only evaluation, therefore, assesses only a fraction of their true capabilities and utility in real-world use cases. We leave it to future iterations

of the framework to incorporate multimodal interactions, which presents a significant research challenge in designing tasks that evaluate not just the quality of outputs in each modality, but also the model's ability to reason and converse coherently across them.

Furthermore, our open-ended conversational design proved to be an imprecise tool for assessing specific, nuanced qualities. As discussed, the high tie rate for *Trust, Ethics & Safety* indicates this methodology does not reliably create a context where such judgments can be made. This limitation points to a clear direction for future research: developing targeted evaluation suites that measure subjective preferences within specialised scenarios. For instance, future studies could use a pairwise comparison framework to evaluate how different models handle sensitive topics, navigate ethical boundaries, or respond to requests for advice in high-stakes domains. Such a focused approach would provide the needed context for users to form discriminative judgments, yielding a much stronger signal than is possible with generic interactions.

## 6 CONCLUSION

The evaluation of large language models requires moving beyond the pursuit of a single, universal score. The HUMAINE framework offers a methodology for this shift, demonstrating that an over-reliance on aggregate scores is insufficient because it obscures critical performance trade-offs, masks demographic blind spots, and misrepresents the utility of different evaluation metrics.

These findings underscore the need for a more nuanced approach to AI development and deployment. For developers, our results highlight the challenge of navigating performance trade-offs across diverse users, rather than simply optimising a singular metric. For organisations, it points towards the importance of a context-aware selection process that aligns a model's specific strengths with their users' needs.

To support this effort, we release our dataset[*] and leaderboard[*] as public resources. Critically, HUMAINE is designed as a living benchmark: the leaderboard is regularly updated as new models are released, ensuring it remains a current reflection of the state-of-the-art. This evaluation approach, which prizes nuance over numbers, is a foundational step towards an evaluation practice that helps catalyse research into AI that is demonstrably equitable, reliable, and genuinely beneficial for the diverse human populations it is meant to serve.

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

# A  STATISTICAL METHODOLOGY: HIERARCHICAL BRADLEY-TERRY-DAVIDSON MODEL

This section formalises the model that turns human A/B/Tie judgements into the leaderboard statistics. It uses a Bradley–Terry–Davidson (BTD) outcome model with hierarchical demographic adjustments and post-stratification to census weights.

## A.1  THE OUTCOME MODEL: PREDICTING A CHOICE

At its core, the model predicts the outcome of a single comparison based on the "latent advantage" ($\eta$) of model $i$ over model $j$. This advantage is the sum of the difference in their baseline skills and the difference in their demographic effects for that specific rater.

The demographic effect for a model ($\Delta u$) is the sum of its adjustments across the rater's age, ethnicity, and political groups:

$$\Delta u_{i,\text{rater}} = u_{i,g_a,k}^{\text{age}} + u_{i,g_e,k}^{\text{eth}} + u_{i,g_p,k}^{\text{pol}} \tag{1}$$

If a rater has multiple labels on an axis, we treat that axis as multi-membership by taking the equal-weight average of the corresponding group adjustments (the weights on that axis sum to 1). If an axis is unobserved for a rater, its contribution is set to 0.

The total advantage, $\eta$, is then:

$$\eta = \underbrace{(\theta_{i,k} - \theta_{j,k})}_{\text{Baseline Skill Difference}} + \alpha \underbrace{(\Delta u_{i,\text{rater}} - \Delta u_{j,\text{rater}})}_{\text{Demographic Effect Difference}} \tag{2}$$

We scale demographic effects by $1/\sqrt{3}$ so that the combined effect of three demographic axes remains on the same scale as a single axis.

Given this advantage $\eta$, the probabilities for each outcome (A wins, Tie, B wins) are calculated using the BTD formula, which includes a per-metric tie propensity $\nu_k > 0$:

$$p_A = \frac{e^{\eta}}{Z}, \qquad p_T = \frac{\nu_k}{Z}, \qquad p_B = \frac{e^{-\eta}}{Z} \qquad \text{where } Z = e^{\eta} + e^{-\eta} + \nu_k \tag{3}$$

## A.2  PRIORS AND LATENT STRUCTURE: HOW PARAMETERS ARE LEARNED

The model's parameters are learned from the data using the following structure and priors:

- **Baseline Skill ($\theta$):** To ensure the skills are identifiable, we enforce a zero-sum constraint for each metric $k$: $\sum_i \theta_{i,k} = 0$
- **Demographic Adjustments ($u$):** The adjustments are learned hierarchically to ensure stability (a technique called partial pooling). For each demographic axis (e.g., age), the adjustments are centred and scaled by a parameter $\tau$ which is learned from the data.

$$u_{i,y,k}^{a} = \left( u_{\text{raw},i,y,k}^{a} - \overline{u_{\text{raw},i,\cdot,k}^{a}} \right) \tau_k^{a} \tag{4}$$

  The raw, unscaled adjustments are drawn from a standard normal distribution, $u_{\text{raw}} \sim N(0,1)$, and the scale parameter $\tau$ (the "volume knob") is drawn from an exponential distribution, $\tau \sim \text{Exponential}(\lambda = 12)$.

## A.3  POPULATION ADJUSTMENT: REFLECTING THE REAL WORLD

After learning the parameters from our participants, we create a population-adjusted skill for each model by taking the expectation of the demographic effects, weighted by census data ($w$). For each posterior draw, this is:

$$\theta_{i,k}^{\text{pop}} = \theta_{i,k} + \alpha \left( \langle w_{\text{age}}, u_{i,\cdot,k}^{\text{age}} \rangle + \langle w_{\text{eth}}, u_{i,\cdot,k}^{\text{eth}} \rangle + \langle w_{\text{pol}}, u_{i,\cdot,k}^{\text{pol}} \rangle \right) \tag{5}$$

Here, $\langle w, u \rangle$ represents the dot product (a weighted average) of the census weights with the model's demographic adjustments for that axis.

## A.4    Scoring and Leaderboard Construction

From the population-adjusted skills, we construct the final leaderboard metrics for each posterior draw:

- The **Expected Points (Winshare)** for model $i$ vs. $j$ is: $\text{EP}_{i \text{ vs } j,k} = p_A + \frac{1}{2}p_T$
- A model's **Score** for that draw is the sum of its EP against all opponents.
- Aggregating these Scores across all posterior draws gives us the final **mean Score**, its uncertainty interval, the **Expected Rank**, and the **P(best)**.

## B    Detailed Methodology

This section provides additional details on the participant experience, data collection procedures, and quality assurance mechanisms employed in the HUMAINE framework.

### B.1    Participant Experience and Interface

Participants accessed the evaluation interface through a web-based platform that presented two AI models side-by-side in an anonymised format (labelled as "Model A" and "Model B"). The interface was designed to minimise cognitive load while ensuring thorough evaluation:

- **Topic Selection:** Participants began by choosing their own conversation topic, eliciting natural engagement and leveraging their interests and domain expertise
- **Synchronised Input:** A single input field sent identical messages to both models simultaneously to help with experimental control. This was a methodological choice made to maintain experimental validity. Allowing independent conversations risked causing diverging conversation trajectories which would have been difficult, and in some instances impossible, to fully to compare.
- **Real-time Responses:** Models responded in parallel with streaming text, allowing participants to observe differences in response speed and style
- **Turn Requirements:** A minimum of 3 conversational turns was enforced before evaluation options became available
- **Evaluation Interface:** After conversation completion, participants rated models across five dimensions using a three-option format (Model A better, Tie, Model B better)

### B.2    Quality Assurance Framework

We developed a quality assurance system which attempted to balance data quality with participant experience:

#### B.2.1    Real-time AI Monitoring

An AI evaluator (`gpt-4o-mini`) analysed messages in real-time to detect:

- Low-effort responses (e.g., single words, repetitive patterns)
- Disjointed conversation flow (unrelated topic jumping)
- Gaming behaviour (attempts to manipulate the system)

When issues were detected, participants received immediate, constructive feedback encouraging higher-quality engagement. The system used a warning-based approach, providing participants opportunities to improve rather than immediate exclusion.

### B.3    Compensation and Participation Structure

- **Fair Compensation:** All participants were compensated at £9 per hour. This matches the platform's (Prolific) recommended rate for ethical research rewards, substantially exceeding the platform's enforced minimum, and meets established ethical standards for research participation

- **Multi-demographic Participation:** Participants qualifying for multiple demographic groups could complete the task once for each relevant group, receiving full compensation for each completion
- **Batch Participation:** Across multiple data collection batches, returning participants received new, randomly assigned model pairs to prevent learning effects
- **Time Investment:** The median task completion time was approximately 15 minutes, including conversation and evaluation phases

## C LLM Judge Implementation Details

### C.1 Metrics and Classifications

To provide a comprehensive and structured characterisation of each conversation, our LLM judge was prompted to generate outputs across three distinct categories: quantitative metrics, categorical classifications, and a detailed qualitative analysis. This multi-faceted approach provided a rich layer of metadata for explaining the patterns observed in human preference data.

**Quantitative Metrics**. The judge assessed each conversation on a 1-5 scale across four independent axes to capture different aspects of the interaction quality and dynamics:

- **Task Complexity:** Measured the cognitive demand of the user's task, ranging from simple fact retrieval (1) to expert-level creative or abstract problem-solving (5).
- **Goal Achievement:** Assessed the degree to which the user's primary objective was accomplished, from complete failure (1) to the model exceeding expectations by providing proactive value (5).
- **User Satisfaction:** Inferred the user's sentiment from their language, ranging from explicit frustration (1) to enthusiastic praise or delight (5).
- **User Engagement:** Quantified the depth of the user's involvement, from a single transactional turn (1) to a deep, collaborative process over an extended dialogue (5).

**Categorical Classifications**. In addition to the quantitative scores, the LLM judge was tasked with classifying each conversation to provide a high-level understanding of its nature.

- **Task Type:** The primary activity the user was engaged in, chosen from a predefined list of 17 types (e.g., *information seeking*, *creative writing*, *coding & debugging*).
- **Domain:** The main subject area of the conversation, chosen from a list of 20 domains (e.g., *technology*, *health & medical*, *finance*).

These classifications allow for large-scale analysis of the types of tasks users naturally bring to LLMs and how preferences may vary across different domains.

### C.2 Analysis Prompt

This section provides the full prompt used for our conversation analysis.

```
Analysis Prompt

You are an expert conversation analyst. Your goal is to score and
categorise a conversation between a user and an AI assistant with
high fidelity, using the entire 1-5 scale for metrics to differentiate
performance. Avoid clustering scores in the middle.

First, you will perform a step-by-step analysis. In a <reasoning>
block, you will provide a brief justification for each metric score,
explicitly referencing the scoring criteria.

Second, after your reasoning, you will provide the final output in
the required JSON format.

CONVERSATION:
{conversation_content}

-------------------------------
```

```
ANALYSIS TASK:

**Step 1: Provide your reasoning within <reasoning> tags.**
For each metric, briefly explain WHY you are choosing a specific
score, referencing the criteria below.

<reasoning>
- **Task Complexity Rationale**: [Explain why the user's task
  deserves a score of 1, 2, 3, 4, or 5 based on the cognitive
  demand. Reference the user's specific requests.]
- **Goal Achievement Rationale**: [Explain the extent to which
  the user's goal was met, partially met, or not met at all.
  Point to evidence in the text.]
- **User Satisfaction Rationale**: [Analyse the user's sentiment.
  Is there explicit praise, frustration, or just neutral acceptance?
  Quote or reference user language.]
- **User Engagement Rationale**: [Describe the depth of the
  interaction. Was it a simple transaction or a deep, collaborative
  exploration? Justify your score.]
</reasoning>

**Step 2: Based on your reasoning above, provide the final JSON.**
Return your analysis in this EXACT JSON format, with no other text
outside the JSON block.

```json
{
  "metrics": {
    "task_complexity": 0,
    "goal_achievement": 0,
    "user_satisfaction": 0,
    "user_engagement": 0
  },
  "categories": {
    "task_type": "information_seeking",
    "domain": "religion",
    "complexity_tier": "medium",
    "engagement_tier": "moderate"
  },
  "detailed_analysis": {
    "conversation_starter": "direct_question",
    "user_initiative": "high",
    "model_proactiveness": "appropriate",
    "goal_achievement_evidence": "User got answer but asked follow-up",
    "primary_user_goal": "Learn about religious practices"
  }
}
```

**CRITICAL SCORING GUIDANCE (for "metrics"):**
- **Use the FULL 1-5 Scale**: You MUST use scores of 1, 2, 4, and 5
  when warranted. A score of 3 is for truly average cases, not a
  default. If a task is a simple fact lookup, it is a 1. Do not
  inflate it.
- **Be a Strict Grader**: Your goal is to create distinctions.
  Scrutinize for flaws and unmet needs.

**SCORING CRITERIA (1-5 Scale, for "metrics"):**

**TASK_COMPLEXITY** - Cognitive demand on the user and model.
1: Simple fact retrieval. Single, unambiguous question.
   E.g., "What is the capital of France?"
2: Simple procedure or explanation. E.g., "How do I boil an egg?"
3: Requires synthesis of a few ideas or multi-step reasoning.
   E.g., "Compare Python and Java for web development."
4: Complex problem-solving or creating a nuanced argument.
   E.g., "Debug this complex code with a race condition."
5: Expert-level, creative, or highly abstract task.
   E.g., "Develop a market entry strategy for South America."
```

```
**GOAL_ACHIEVEMENT** - Was the user's objective accomplished?
1: Goal completely failed. User is explicit about failure or abandons.
2: Goal mostly failed. Core need is unmet.
3: Goal partially met. Main question answered, but user needs follow-up.
4: Goal fully met. User's stated goal is clearly accomplished.
5: Goal exceeded. Model was proactive and provided value beyond request.

**USER_SATISFACTION** - User's sentiment about the interaction.
1: Explicit frustration or anger.
2: Implicit frustration, impatience, or mild disappointment.
3: Neutral. Purely transactional.
4: Positive. User says "thanks," "perfect," or other positive indicators.
5: Enthusiastic. User expresses strong praise or delight.

**USER_ENGAGEMENT** - Depth of user's active involvement.
1: Single turn. One question, one answer.
2: Minimal follow-up. One or two simple clarifying questions.
3: Moderate exploration. User asks several related questions.
4: Active collaboration. User refines prompts, challenges the model.
5: Deep co-creation. Extended dialogue building complex understanding.

**CATEGORIES (for "categories"):**

**TASK_TYPE**: Choose from: information_seeking, technical_assistance,
creative_writing, problem_solving, tutoring_explanation, brainstorming,
research, writing_help, coding_debugging, analysis_review,
project_planning, personal_advice, social_conversation, content_creation,
learning_education, decision_support, comparison_analysis.

**DOMAIN**: Choose from: technology, science, business, education,
health_medical, creative_arts, finance, law_legal, cooking_food,
travel, relationships, career_professional, academic_research,
programming, design, entertainment, sports, religion_philosophy,
history, language_linguistics.

**TIER CLASSIFICATIONS**:
- **COMPLEXITY_TIER**: low (scores 1-2), medium (3), high (4-5)
- **ENGAGEMENT_TIER**: low (scores 1-2), moderate (3), high (4-5)
```

## C.3 IMPLEMENTATION NOTES

Conversations exceeding 50 turns were truncated to prevent context window issues. The prompt employs a two-stage reasoning approach where the LLM must first articulate its reasoning for each score before generating the final JSON output. This design reduces anchoring bias, the tendency for the model to commit to an initial score and then rationalise it post-hoc, by ensuring scores are grounded in explicit reasoning rather than intuition. Failed API calls or JSON parsing errors were logged and excluded from analysis.

## D DEMOGRAPHIC INTERACTION ANALYSIS

This section presents a detailed empirical analysis of interaction effects between demographic dimensions to assess whether preference patterns arise from combinations of demographic factors (e.g., age × politics) beyond their individual additive contributions.

### D.1 MOTIVATION AND APPROACH

The hierarchical BTD model employed in the main analysis assumes an additive structure for demographic effects. Under this model, a participant's preference is decomposed as:

$$\text{preference} = \text{baseline} + \text{age effect} + \text{ethnicity effect} + \text{politics effect} \tag{6}$$

This assumes that the effect of being in a particular age group (e.g., 55+) does not systematically vary depending on one's political affiliation. While the hierarchical structure with partial pooling

allows the model to share information across all demographic combinations and learn group-specific patterns, it does not explicitly model interaction terms.

We conducted a direct analysis of tie rates across all demographic combinations to empirically assess the validity of this additive approximation. Tie rates serve as a proxy for preference uncertainty and provide an interpretable metric for examining whether certain demographic combinations exhibit systematically different patterns than would be predicted by an additive model.

### D.2    METHODOLOGY

We employed a standard two-way ANOVA decomposition to partition observed tie rates into main effects and interaction effects. For each pair of demographic dimensions (age × politics, age × ethnicity, politics × ethnicity), we computed:

1. **Grand mean** ($\mu$): The overall average tie rate across all demographic groups
2. **Row effects** ($\alpha_i$): The deviation of each row group's marginal mean from the grand mean
3. **Column effects** ($\beta_j$): The deviation of each column group's marginal mean from the grand mean
4. **Expected values under additive model**: $E[y_{ij}] = \mu + \alpha_i + \beta_j$
5. **Interaction effects**: $\gamma_{ij} = y_{ij}^{\text{observed}} - E[y_{ij}]$

The interaction effects ($\gamma_{ij}$) represent deviations from the additive model (the portion of the observed tie rate that cannot be explained by the main effects alone). A large interaction effect indicates that the combination of being in row group $i$ and column group $j$ produces a tie rate that is substantially different from what would be predicted by simply adding their individual effects.

### D.3    RESULTS

We conducted this analysis separately for the US and UK populations to account for their different demographic compositions and political systems. Each decomposition shows three panels: (left) observed tie rates, (middle) expected tie rates under an additive model, and (right) interaction effects in percentage points (the difference between observed and expected values).

#### D.3.1    UNITED STATES RESULTS

Figures 5, 6, and 7 show the decompositions for the US population across the three demographic pairs.

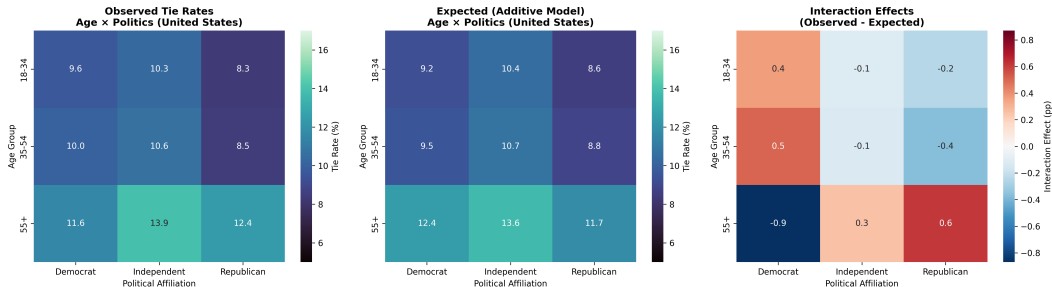

Figure 5: Decomposition of tie rates for Age × Politics (US). Left: Observed tie rates. Middle: Expected tie rates under additive model (grand mean + row effect + col effect). Right: Interaction effects in percentage points (observed - expected).

The US data shows that interaction effects explain 7-41% of variance in tie rates. Age × politics shows minimal interaction (7.0%), indicating that age and political affiliation have largely independent effects. In contrast, age × ethnicity (41.3%) and politics × ethnicity (40.7%) show substantial interactions, suggesting these demographic factors combine in more complex ways. The maximum interaction effect is 2.0 percentage points. All US demographic cells have at least 170 observations, providing stable estimates.

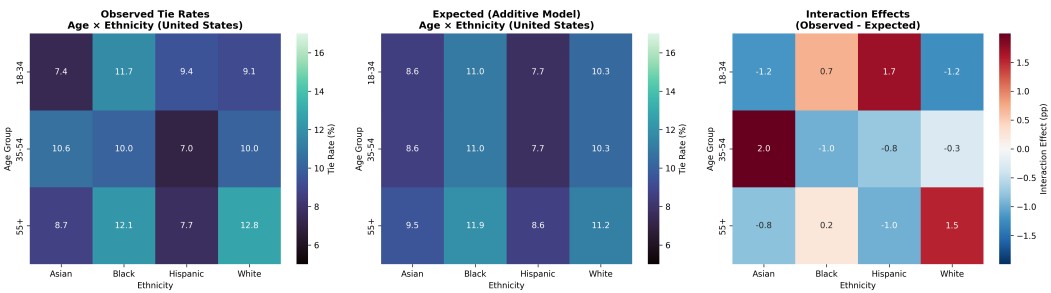

Figure 6: Decomposition of tie rates for Age × Ethnicity (US). Format as in Figure 5.

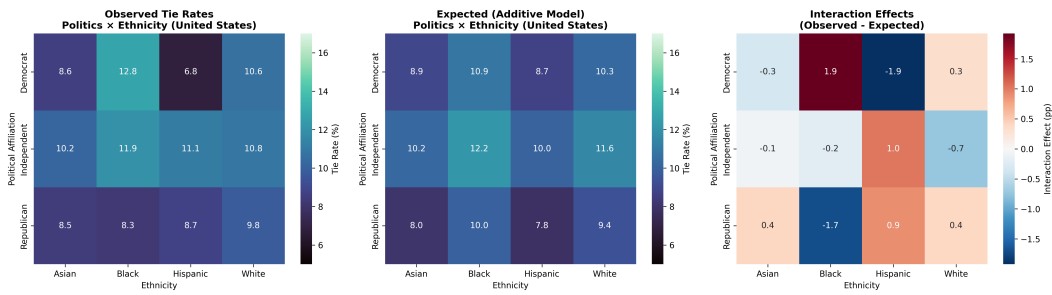

Figure 7: Decomposition of tie rates for Politics × Ethnicity (US). Format as in Figure 5.

### D.3.2 UNITED KINGDOM RESULTS

Figures 8, 9, and 10 show the decompositions for the UK population.

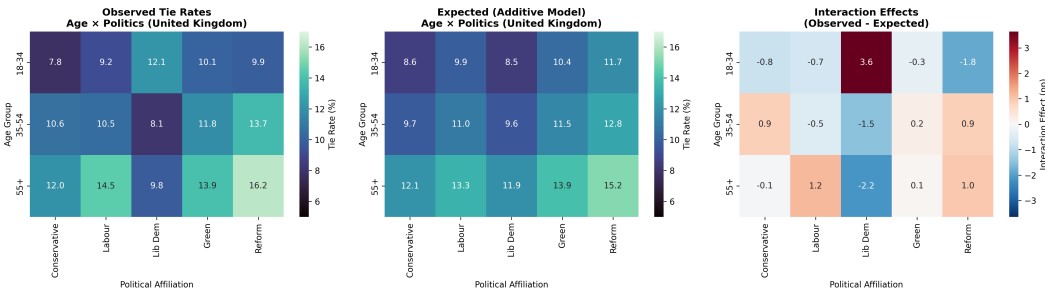

Figure 8: Decomposition of tie rates for Age × Politics (UK). Format as in Figure 5.

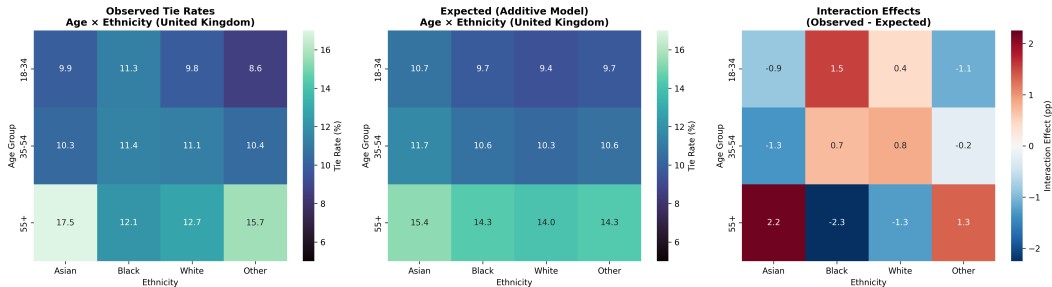

Figure 9: Decomposition of tie rates for Age × Ethnicity (UK). Format as in Figure 5.

The UK data shows higher interaction effects overall, explaining 29-50% of variance. Unlike the US, age × politics shows substantial interaction (35.2%), suggesting the relationship between age and political affiliation operates differently in the UK. Politics × ethnicity shows the strongest interaction

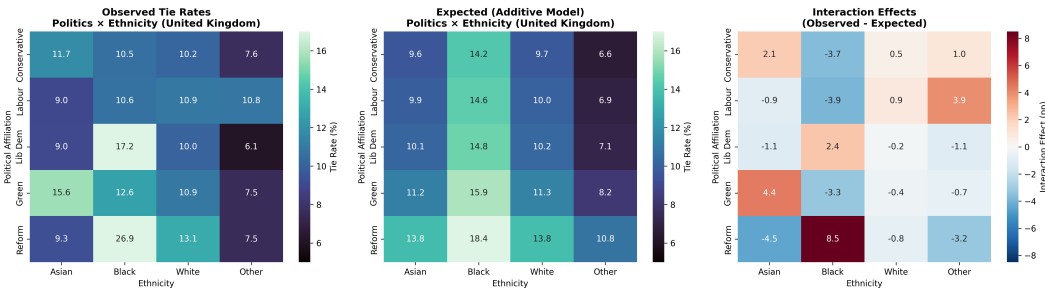

Figure 10: Decomposition of tie rates for Politics × Ethnicity (UK). Format as in Figure 5.

(49.7%) with a maximum effect of 8.5 percentage points. However, some UK demographic cells have sparse sample sizes (e.g., Black × Reform UK: n=26, Asian × Reform UK: n=54), which may contribute to larger estimated interaction effects due to higher sampling variability.

Table 1 summarises the magnitude and variance explained by interaction effects across all three demographic pairs for both the US and UK.

Table 1: Summary of interaction effects across demographic pairs

| Country | Demographic Pair | Variance from Interaction | Max Interaction | Mean Interaction |
|---|---|---|---|---|
| US | Age × Politics | 7.0% | 0.009 | 0.004 |
| US | Age × Ethnicity | 41.3% | 0.020 | 0.010 |
| US | Politics × Ethnicity | 40.7% | 0.019 | 0.008 |
| UK | Age × Politics | 35.2% | 0.036 | 0.010 |
| UK | Age × Ethnicity | 28.9% | 0.023 | 0.012 |
| UK | Politics × Ethnicity | 49.7% | 0.085 | 0.024 |

### D.4 INTERPRETATION

**Cross-country comparison.** The most striking finding is the difference in age × politics interactions between the US (7.0%) and UK (35.2%). In the US, age and political affiliation have largely independent effects on tie rates, suggesting these dimensions operate as separate factors. In the UK, however, these dimensions interact substantially, suggesting that political identity and generational cohorts align differently across the two countries' political systems. This difference may reflect the UK's more fragmented multi-party system compared to the US two-party structure.

Both countries show substantial politics × ethnicity interactions, though the UK effect (49.7%) exceeds the US (40.7%). This suggests that the intersection of political affiliation and ethnicity creates unique preference patterns that cannot be captured by considering these dimensions independently. The systematic deviations from the additive model indicate that certain demographic combinations produce tie rates that differ meaningfully from what would be predicted by simply adding their individual effects.

### D.5 IMPLICATIONS FOR MODEL DESIGN

Given that interactions explain 7-50% of variance depending on the demographic pair and country, one might ask whether explicit interaction terms should be included in the model. We chose the additive approximation for the following reasons:

1. **Sample size**: Two-way interactions have median sample sizes of 400-650 observations per cell, which is theoretically sufficient for hierarchical modelling. However, three-way interactions (age × ethnicity × politics) have median sample sizes of only 105 observations per cell, with 47% of cells having fewer than 100 observations. Modelling three-way interactions would risk severe overfitting.

2. **Interpretability**: The additive structure allows clear interpretation of demographic effects as deviations from a baseline, which aligns with our research goal of understanding which demographic dimensions drive preference heterogeneity. Adding 30+ two-way interaction parameters would substantially complicate interpretation.

3. **Model complexity**: While two-way interactions are feasible for some pairs (particularly age × ethnicity and politics × ethnicity), validating and interpreting a model with explicit interaction terms requires careful specification of prior structures and model comparison. We prioritised the simpler additive model to maintain transparency and avoid the risk of overfitting to noise in demographic combinations.

We acknowledge this as a methodological trade-off: the additive model captures the dominant structure for age × politics (93% of variance from main effects) but may underfit age × ethnicity and politics × ethnicity where interactions are more substantial. The hierarchical structure absorbs some demographic-specific interaction patterns through partial pooling rather than explicitly parameterising them. Future work could explore hierarchical models with two-way interaction terms using appropriate regularisation.

# E    PILOT STUDY: DERIVATION OF EVALUATION DIMENSIONS

We conducted an initial user experience study to develop a comprehensive evaluation framework that captures multiple aspects of how users perceive and rate LLM performance in everyday use cases. This pilot informed the dimensional structure used in the main HUMAINE benchmark.

## E.1    INITIAL STUDY DESIGN

**Participants and Tasks.** We recruited 514 participants demographically representative of the US population (stratified by age, sex, ethnicity, political affiliation). Each participant completed six everyday tasks spanning creative, practical, and analytical use cases: (1) following up on a job application, (2) planning weekly meals with dietary restrictions, (3) creating a European travel itinerary, (4) understanding complex topics (day trading), (5) generating creative gift ideas, and (6) making product purchase decisions. Each task was completed with a different LLM (o1, gpt-4o, claude-3.7, gemini-2-flash, llama-3.1-405b, deepseek-r1), with random assignment and ordering. Participants exchanged at least 4 messages per conversation.

**Initial Evaluation Framework.** Participants rated their experience along 30 dimensions organised in 7 high-level categories (rated on 1-7 Likert scales) and 21 nested more specific metrics (rated on 1-5 scales), and 2 additional standalone metrics:

1. **Helpfulness:** Effectiveness, Comprehensiveness, Usefulness
2. **Communication:** Tone and Language Style, Conversation Flow, Detail and Technical Language
3. **Understanding:** Accuracy, Context Memory, Intuitiveness
4. **Adaptiveness:** Flexibility, Clarity, Conversation Building
5. **Trustworthiness:** Consistency, Confidence, Transparency
6. **Personality:** Personality Consistency, Distinct Personality, Honesty/Empathy/Fairness
7. **Background and Culture:** Ethical Alignment, Cultural Awareness, Bias and Stereotypes
8. **Additional metrics:** Repeat Usage, Speed Perception

Data was processed through multiple regression with poststratification (MRP) models to create nationally representative estimates, where all of the model estimations were parametrically bootstrapped (N=1000) to account for uncertainty.

## E.2    FACTOR ANALYSIS PROCESS

To derive a more parsimonious set of evaluation dimensions, we conducted an extensive factor analysis on the 30 collected metrics. Pre-analysis checks confirmed data suitability via the Kaiser-Meyer-Olkin measure (KMO = 0.931, excellent) and Bartlett's Test of Sphericity ($p < 0.001$), indicating strong correlations between many original categories and the presence of underlying latent constructs. The scree plot (Figure 11) indicated 4 factors with eigenvalues greater than 1.

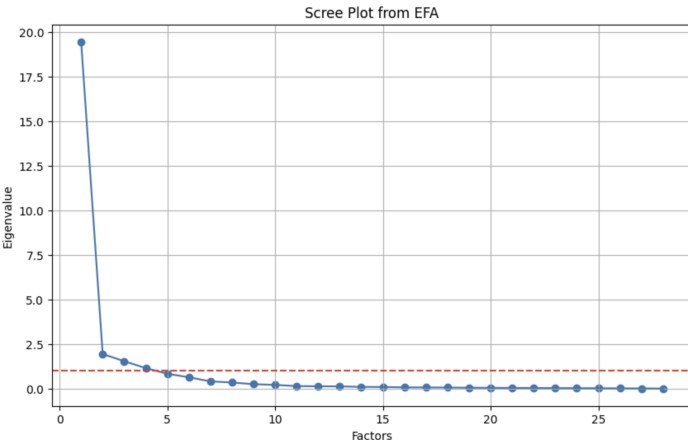

Figure 11: Scree plot showing eigenvalues for each factor, indicating 4 factors with eigenvalues ¿ 1.

**Exploratory Factor Analysis.** We conducted iterative EFA with promax rotation, testing 3-, 4-, and 5-factor models. The 3-factor and 5-factor models exhibited Heywood cases (communalities > 1.0), indicating over-extraction or problematic variables. The 4-factor model showed improved stability.

**Confirmatory Factor Analysis.** CFA attempts to validate an initial 5-factor theoretical model (Task Effectiveness, Communication Quality, Reasoning & Understanding, Adaptiveness, Trust & Safety) revealed extreme multicollinearity. Standardised inter-factor correlations exceeded 1.0 (e.g., Task Effectiveness & Reasoning = 1.006), indicating these constructs were empirically indistinguishable.

### E.3 FINAL FACTOR SELECTION

Based on the factor collapse patterns, we derived a refined set of 4 metrics, which provided a good balance between empirical support and conceptual granularity. All four metrics demonstrated excellent internal consistency (Cronbach's alpha):

1. **Core Task Performance & Reasoning** (8 items, $\alpha = 0.969$)
2. **Interaction Fluidity & Adaptiveness** (6 items, $\alpha = 0.960$)
3. **Communication Style & Presentation** (6 items, $\alpha = 0.893$)
4. **Trust, Ethics & Safety** (8 items, $\alpha = 0.925$)

This analysis revealed that participants do not seem to make fine-grained distinctions between many of the conceptual aspects. Task Effectiveness, Reasoning & Understanding, and Adaptiveness combined seem to form a highly inter-related construct related to overall performance and cognitive capability. Trust & Safety emerged as a relatively more distinct construct, though still correlated with general performance. These four dimensions, validated through this pilot study with Cronbach's alpha > 0.89 for all metrics, formed the basis for the evaluation metrics in the main HUMAINE benchmark.

