# OpenReview forum: "Unpacking Human Preference for LLMs: Demographically Aware Evaluation with the HUMAINE Framework"
_ICLR.cc/2026/Conference — ICLR 2026 Poster_

### Official Review · Reviewer_3vQb · 2025-10-30

**Soundness:** 3
**Presentation:** 2
**Contribution:** 3
**Rating:** 6
**Confidence:** 4

**Summary:**

This paper proposes DIVERSE, a demographically aware, multidimensional framework for human evaluation of LLMs. The authors collect large-scale, multi-turn pairwise comparisons from stratified US/UK participants and analyze results with a hierarchical Bayesian Bradley–Terry–Davidson model combined with census post-stratification. They report a clear performance hierarchy among contemporary models, with one model consistently leading, but show that preferences vary meaningfully across demographic groups—especially by age—altering perceived ranks. They also find that evaluation dimensions differ markedly in how decisively users can discriminate between models, with holistic judgments being more decisive than safety-oriented ones. The work advocates moving beyond single-metric leaderboards toward nuanced, population-aware assessment.

**Strengths:**

1. This paper surveys a large, demographically stratified US/UK population across many groups, collects extensive multi-turn data, and commits to releasing it—providing a high-quality community resource.

2. The authors use parallel pairwise dialogues and a hierarchical Bayesian BTD model with tie handling and post-stratification, explicitly modeling demographic heterogeneity to produce uncertainty-aware, multidimensional rankings.

3. The paper also provides actionable, empirically grounded insights. It shows that “overall” judgments align closely with core task/reasoning while rankings shift on communication and safety; reveals systematic age-driven preference differences with older users more tie-prone; and demonstrates large disparities in metric discriminability. These discoveries can guide audience- and task-specific model selection and alignment.

**Weaknesses:**

1. The participant pool, though large, is limited to US/UK and underrepresents broader cultural and socioeconomic contexts (e.g., non‑Anglophone regions, Global South populations, rural communities, low digital‑literacy users, non‑binary identities, and diverse education/income strata).

2. The paper reports five dimensions but offers little analysis linking the four sub-dimensions to the overall score or unpacking how conversational features (topic, complexity, safety triggers) map to those judgments; the chosen sub-dimensions may also miss facets like creativity, empathy, humor, and long-horizon reliability.

3. Data arise from a study interface rather than organic, in-the-wild use, so prompts and stakes may diverge from real workflows, creating ecological bias; in addition, compared with standard automated benchmarks, this framework is harder to deploy and slower to collect, and the paper could better justify the return on that extra effort.

4. The related work section could be broadened to cover more papers on human preference variability and pitfalls in human feedback (e.g., “Human Feedback is Not a Gold Standard,” ICLR 2024; “Towards Understanding Sycophancy in Language Models,” ICLR 2024; “Dissecting Human and LLM Preferences,” ACL 2024).

**Questions:**

See Weaknesses.

---

> ### Author Response · Authors · 2025-11-18
> **Geographic Limitations, Ecological Validity, and Preference Variability Literature**
>
> Thank you for the thoughtful and balanced assessment!
>
> **Interactive Leaderboard:** https://diverse-leaderboard-565860601788.europe-west1.run.app/ (anonymised for review)
>
> **Released Datasets:**
> - Preference feedback dataset: https://drive.google.com/file/d/1pC8ZQ7AnjZti4ibcXsspSo15OC9Wq2m2/view?usp=sharing
> - Conversation metadata dataset: https://drive.google.com/file/d/1XKwsOtwbM9W4h6Ow58FhwtA7sEHVUIAP/view?usp=sharing
>
> Non-anonymised versions will be made available upon publication.
>
> ## Weakness 1: Geographic and Demographic Limitation
>
> You're correct that our geographic scope is limited to US/UK populations. Our contribution addresses within-population sampling bias through stratified sampling and post-stratification to census data, making sure that results represent actual US/UK population preferences rather than self-selected samples.
>
> We stratified by three axes: age, ethnicity, and political affiliation. This doesn't mean our dataset lacks other demographic dimensions, our recruitment naturally includes diversity in education levels, income strata, and gender identities. We simply didn't stratify by these variables. Our stratification choices were guided by: (1) prior evidence these factors influence AI preferences (Santurkar et al., 2023), (2) census data availability for post-stratification, and (3) practical feasibility, where each additional axis exponentially increases required sample size.
>
> It is a good point that non-Anglophone regions, Global South populations, rural communities, and low digital-literacy users are underrepresented. We've now made the geographic limitation more prominent in the Abstract and Introduction, explicitly stating findings apply to US/UK English-speaking populations, and reframed the contribution to position DIVERSE as demonstrating a methodology for demographically-aware evaluation with an initial US/UK implementation.
>
> Expanding to other countries, languages, and broader demographic stratification is a priority for future work.
>
> ## Weakness 2: Limited Analysis of Sub-Dimensions and Missing Dimensions
>
> The interactive leaderboard (linked above) provides analysis of how sub-dimensions relate to Overall Winner and to conversational features. You can explore these relationships in the "Conversation Analysis" tab.
>
> Our final dimensions were determined empirically through factor analysis of pilot data on ~30 initial metrics (Appendix E), necessary to avoid participant cognitive overload and eliminate redundant correlated dimensions.
>
> We acknowledge that qualities like creativity and empathy aren't explicitly isolated. They may be partially captured (e.g., creative answers might score well on Communication Style and Task Performance), but our methodology may underweight their importance in contexts like creative writing or therapeutic chatbots. Long-horizon reliability and safety were out of scope but are valuable future directions. We've expanded our discussion of these limitations in Section 5.1.
>
> ## Weakness 3: Ecological Validity and Deployment Difficulty
>
> We acknowledge our study interface isn't "in-the-wild" LLM use. But perfect ecological validity in human preference evaluation is practically unattainable -- no design can simultaneously observe genuine day-to-day interactions while collecting systematic preference judgments. The very act of measuring preferences introduces a study context.
>
> Our design prioritises a different form of ecological validity: participants choose their own topics and interact naturally over multiple turns, striking a balance between experimental control and naturalistic interaction.
>
> On deployment difficulty: yes, our framework requires more resources than automated benchmarks. But human preference evaluation and automated benchmarks measure fundamentally different things—technical capability versus subjective user experience.
>
> DIVERSE provides what automated benchmarks can't: subjective qualities that require human judgment, preference heterogeneity that's critical for deployment decisions, real-world task distribution, and dimensional trade-offs that aggregate scores mask.
>
> The overhead is the necessary cost of measuring phenomena automated approaches can't capture. The question isn't whether human preference data is worth collecting, but whether it's collected rigorously. Our released framework and datasets reduce the barrier for others to do the same.
>
> ## Weakness 4: Related Work - Preference Variability Literature
>
> Thank you for the specific citations. We have expanded our related work section to better situate our contribution within the literature on human preference heterogeneity and the complexities of human feedback. We have added discussion of relevant work including "Human Feedback is Not a Gold Standard" (ICLR 2024) and "Dissecting Human and LLM Preferences" (ACL 2024), which directly address preference variability and connect to our core findings.

---

> > ### Comment · Reviewer_3vQb · 2025-11-27
> > **Response to rebuttal**
> >
> > Thanks for the responses and the additional results/explanations/public websites. I think they have addressed my concerns and this paper makes a decent contribution to the community. I will maintain a positive attitude toward this paper and think it deserves acceptance. I would give this paper a rating of 7 if we had such an option.

---

### Official Review · Reviewer_HzhH · 2025-10-31

**Soundness:** 4
**Presentation:** 4
**Contribution:** 4
**Rating:** 10
**Confidence:** 3

**Summary:**

This paper presents an exciting ongoing effort to collect real-world preference data from people across diverse demographics. Much of this information was not available in other public datasets. Having a dataset with detailed breakdowns of diverse human preferences will be very valuable to the scientific community in the long run.

**Strengths:**

* The pairwise preference dataset collection is very well executed with adaptive sampling to select the most uncertain pair and quality controls with GPT-4o-mini as the LLM judge.

* The dataset is collected in large-scale with 106K pairwise data from 21K participants across 27 LLMs and will be made publicly available.

* This paper also introduces a novel evaluation framework based on the hierarchical BTD model. This methodology contribution is just as important as the dataset contribution.

**Weaknesses:**

* It seems that participants are instructed to have at least 3 turns in the study. It will be great to have a deeper analysis/discussion on how the depth of a conversation impact quality/model performance.

* Another dimension that authors could consider is break down by tasks. One may hypothesize that preferences may vary across both tasks and demographics.

* I highly suggest the authors to make this living benchmark and ongoing dataset release.

**Questions:**

N/A

---

> ### Author Response · Authors · 2025-11-18
> **Conversation Analysis and Living Benchmark Commitment**
>
> Thank you for the support and for recognising the value in both the dataset and methodological contributions.
>
> **Interactive Leaderboard:** https://diverse-leaderboard-565860601788.europe-west1.run.app/ (anonymised for review)
>
> **Released Datasets:**
> - Preference feedback dataset: https://drive.google.com/file/d/1pC8ZQ7AnjZti4ibcXsspSo15OC9Wq2m2/view?usp=sharing
> - Conversation metadata dataset: https://drive.google.com/file/d/1XKwsOtwbM9W4h6Ow58FhwtA7sEHVUIAP/view?usp=sharing
>
> Non-anonymised versions of the datasets and the leaderboard will be made available upon publication.
>
> ## Weakness 1: Conversation Depth Analysis
>
> We looked into how conversation depth relates to evaluation outcomes. Our dataset shows a median conversation length of 6 turns. Analysing goal achievement scores across conversation lengths reveals essentially no correlation (r = -.004), indicating that conversation quality and user satisfaction remain consistent regardless of length. This suggests participants form reliable preferences even in shorter interactions.
>
> There are many worthwhile analyses of this type that could be conducted. The datasets and interactive leaderboard linked above are designed to support researchers in exploring these questions, including conversation-level patterns across demographics, task types, and evaluation dimensions.
>
> ## Weakness 2: Task Breakdown Analysis
>
> The interactive leaderboard (linked above) provides task-level analysis. Navigate to the "Conversation Analysis" tab to explore breakdowns by task type, topic, task complexity and engagement.
>
> We did not incorporate task type into our primary hierarchical model for two reasons:
>
> **Firstly, ecological validity:** We prioritised naturalistic interactions over controlled task assignment. Prescriptive prompts risk constraining evaluation to researcher-anticipated use cases and requiring participants to engage with topics outside their domain knowledge, which would compromise judgment reliability. Our design captures the natural heterogeneity of real world interactions across actual use cases.
>
> **Secondly, reliability constraints:** Task categorisation relies on LLM-based post-hoc classification (Section 3.4.2). While useful for exploratory analysis, LLM classification lacks the reliability required for formal covariate inclusion in our primary model (Schroeder & Wood-Doughty, 2024; Shi et al., 2024; Wataoka et al., 2024). Additionally, the resulting task distribution is highly imbalanced (information seeking: 71.5%, technical assistance: 2.4%, creative writing: 1.3%), making task-stratified analysis unreliable.
>
> ## Weakness 3: Living Benchmark & Ongoing Dataset Release
>
> We strongly agree and are fully committed to this. The interactive leaderboard is deployed and publicly accessible at the link above. We are actively maintaining DIVERSE as a living benchmark, for example, we enrolled Claude Sonnet 4.5 soon after its release in October.

---

### Official Review · Reviewer_Ba5U · 2025-11-04

**Soundness:** 2
**Presentation:** 2
**Contribution:** 2
**Rating:** 2
**Confidence:** 5

**Summary:**

This paper presents a large-scale dataset capturing human preferences over pairs of language model outputs along five evaluation dimensions, stratified by three demographic attributes (age, ethnicity, and politics), covering 27 models. The authors analyze these preferences using a hierarchical Bradley–Terry–Davidson model to learn demographic-specific adjustments and quantify preference variation. Their analysis reveals (1) the overall ranking of models, (2) age as the most divergent demographic axis, (3) variation in rankings across evaluation dimensions, and (4) differences in human tie rates across dimensions, indicating varying decisiveness among annotators.

**Strengths:**

- Well-motivated and timely research direction. The paper tackles an increasingly critical issue in LLM evaluation: how human preferences differ across demographic groups and qualitative dimensions. It contributes a more nuanced understanding of model performance evaluation.
- Valuable dataset contribution. The dataset could be a useful addition to the community: it is large-scale, multi-turn, demographically stratified, and spans a broad set of 27 LLMs. Such a dataset can serve as an important benchmark for studying model preference heterogeneity and subjective quality dimensions beyond conventional accuracy-based benchmark metrics.
- Methodological rigor in modeling human preferences: The paper goes beyond simple aggregation of crowd judgments by employing a hierarchical Bradley–Terry–Davidson model. This allows the authors to systematically estimate demographic-specific adjustments and quantify preference variation.

**Weaknesses:**

- Insufficient alignment between the stated problem and the proposed solution. The paper’s introduction highlights two major issues: (1) the dominance of single-metric evaluation and (2) the neglect of subjectivity. However, the proposed solution, evaluating models on five dimensions across demographic strata, resembles running multiple benchmarks, each focusing on a different aspect. While this is a meaningful improvement, it does not fully capture the “subjectivity” aspect claimed in the introduction, since subjectivity extends beyond demographic factors and fixed evaluation axes.
- Brief and unclear data description. Section 3.2 should provide a more comprehensive account of the dataset. For example, it mentions that “each message sent by the participant was delivered to both models simultaneously” and that “a minimum of 3 conversational turns were required,” but it is unclear how the same message could be reused across multiple turns, given the dependency on previous model responses. The paper also states that participants were free to select topics—what kinds of topics did they typically choose? Furthermore, while gpt-4o-mini was used to flag “low-effort inputs,” the definition and examples of such inputs are not provided. Given that the dataset is the paper’s primary contribution, the absence of even a single example conversation or basic statistics (e.g., number of evaluations per stratum, average conversation length) makes it difficult to grasp the dataset’s structure and richness.
- Overgeneralization and overstatement of findings. The paper emphasizes that “age is the most significant demographic factor driving preference heterogeneity,” but this conclusion holds only among the three studied axes (age, ethnicity, politics). The claim should be scoped accordingly. Similarly, while the paper asserts that “a model’s competitive standing can change dramatically depending on the evaluation lens,” Figure 3 shows high correlation across dimensions, implying that rankings are fairly consistent rather than dramatically shifting. These inconsistencies weaken the strength of the argument.
- Weak overall contribution and presentation quality. The paper feels stretched for a full-length submission. For example, Section 3.4.2 describes an “LLM judge for conversational analysis,” but this analysis is absent from the results. The discussion section repeats much of the results content rather than extending it, which makes it redundant. Some metrics, such as “average rank shifts” (Section 4.2) and “tie percentage” (Section 4.4), are insufficiently defined. These issues collectively reduce the paper’s overall impact and quality.

**Questions:**

- The compensation rate of £9/hr is described as "recommended" (line 157), yet this falls below the UK minimum wage of £12.21/hr for workers aged 21 and over. Could you clarify how this rate was determined and whether it meets ethical guidelines for research compensation?
- The paper repeatedly refers to DIVERSE as a "living benchmark" with "continuous evaluation" and claims to "continuously add new models and update rankings." Is there an active platform where ongoing evaluation is happening? If not, these claims seem misleading and should be clarified or removed.
- The four evaluation dimensions are stated to derive from a pilot study using factor analysis (Section 3.3), but no details about this study are provided. Given these dimensions are central to your analysis, could you describe the pilot study methodology, sample size, and how factor analysis led to these specific dimensions?

**Details Of Ethics Concerns:**

The paper reports that they compensated US and UK annotators at the rate of £9/hr (line 157) which falls below the UK minimum wage of £12.21/hr for workers aged 21 and over. Would benefit to check with authors to make sure it meets ethical guidelines for research compensation.

---

> ### Author Response · Authors · 2025-11-18
> **Methodological Clarifications, Compensation Ethics, and Pilot Study Details**
>
> Thank you for the thoughtful and detailed review. We've now made the following directly available:
>
> **Interactive Leaderboard:** https://diverse-leaderboard-565860601788.europe-west1.run.app/
> (Navigate to "Conversation Analysis" > "Data Deep Dive" for task types, topics, complexity, and engagement metrics)
>
> **Datasets:**
> - Preference feedback: https://drive.google.com/file/d/1pC8ZQ7AnjZti4ibcXsspSo15OC9Wq2m2/view?usp=sharing
> - Conversation metadata: https://drive.google.com/file/d/1XKwsOtwbM9W4h6Ow58FhwtA7sEHVUIAP/view?usp=sharing
>
> These resources address your concerns about dataset transparency, topic distribution, and conversation statistics.
>
> ## Weakness 1: Alignment Between Problem and Solution
>
> We agree our framing could be clearer. We now acknowledge in our limitations (Section 5.1) that our framework is not exhaustive.
>
> That said, we believe characterising our approach as "running multiple benchmarks" understates the contribution. DIVERSE captures subjectivity along three critical axes: demographic variation (different users have different preferences), dimensional variation (quality is multi-faceted), and contextual variation (what users want depends on their use case). The core contribution is jointly modelling how preferences vary across all three simultaneously. Our open-ended design resulted in conversations spanning 41 domains, and combined with demographic stratification and dimensional evaluation, we can measure genuine preference heterogeneity across populations, use cases, and quality dimensions.
>
> ## Weakness 2: Data Description and Examples
>
> We've now expanded Section 3.2 with detailed data description and added Appendix B1 on experimental mechanics. The datasets and interactive leaderboard (linked above) provide full conversation statistics, task distributions, and metadata for direct examination.
>
> ## Weakness 3: Overgeneralisation and Overstatement
>
> **Age finding:** You're right that greater clarity is needed. We've revised to explicitly scope the claim: "Among the three demographic axes we studied (age, ethnicity, political affiliation), age emerged as the most significant driver of preference heterogeneity" in Results, Discussion, and Limitations.
>
> **"Dramatic" ranking shifts:** We've revised "dramatic" to "meaningful." While high rank correlations show strong models are generally strong across dimensions, mid-tier models exhibit mean rank ranges of 5.7 positions, revealing dimensional specialisation that matters for context-specific use cases.
>
> ## Weakness 4: Weak Contribution and Presentation Quality
>
> We've made targeted revisions:
>
> **LLM judge analysis in Results:** Added Section 4.5 with descriptive statistics (task types, topics, complexity, engagement) to show conversation breadth.
>
> **Metric definitions:** Defined "average rank shift" (Section 4.2) and "tie percentage/rate" (Section 4.4) when first introduced.
>
> **Limitations expanded:** Clarified our demographic stratification focused on three axes, acknowledging other factors may reveal additional preference heterogeneity.
>
> ## Question 1: Compensation Ethics
>
> We appreciate the concern and want to clarify the ethical framework governing research participation.
>
> The £9/hour rate refers to payment for voluntary participation in a research study, not an employment relationship. Research participation doesn't constitute employment, so statutory minimum wage legislation doesn't apply.
>
> **Our rate of £9/hour exactly matches Prolific's recommended minimum** (Prolific Payment Principles, 2024). Prolific enforces a £6/hour minimum but recommends £9/hour for ethical research recruitment. Our rate substantially exceeds the enforced minimum and meets the platform's ethical standards.
>
> Research ethics guidelines focus on ensuring participants are treated fairly, not exploited, and not unduly induced (British Psychological Society, 2021; ESRC Framework, 2023). UK university guidance recommends £6–10/hour for online studies. Our rate falls within this range and aligns with widely accepted standards.
>
> ## Question 2: "Living Benchmark" Status
>
> The leaderboard is deployed at https://diverse-leaderboard-565860601788.europe-west1.run.app/. We update as new models are released (e.g., Claude Sonnet 4.5 soon after release) and plan rapid onboarding (<1 week) for major releases. We've clarified in the paper that the framework is designed for extensibility to additional geographies, demographics, and models.
>
> ## Question 3: Pilot Study Details
>
> We've added a complete Pilot Study appendix (Appendix E) with full methodological transparency. We derived our evaluation dimensions through factor analysis of ~30 Likert-scale items from 514 participants. The process involved: pre-analysis checks (KMO = 0.931), iterative Exploratory Factor Analysis, Confirmatory Factor Analysis, and derivation of a final 4-factor solution with excellent internal consistency (Cronbach's alpha 0.893-0.969). The appendix includes complete methodology, scree plots, and results.

---

### Official Review · Reviewer_n89e · 2025-11-05

**Soundness:** 2
**Presentation:** 3
**Contribution:** 4
**Rating:** 6
**Confidence:** 4

**Summary:**

The paper introduces DIVERSE, a large-scale framework for LLM evaluation that is demographically aware, multi-dimensional, and human-AI interaction focused. The study collects ~110K pairwise comparisons of 27 models from ~21K participants across 22 different demographic groups in the US/UK. Using a hierarchical bayesian bradley-terry model with tie handling and demographic effect adjustments, they found that Google's gemini-2.5-pro achieved the top performance in most dimensional analyses. Lastly the authors emphasized that the context aware design of LLM evaluation is highly essential to accurately measure a model's capacity with user needs.

**Strengths:**

- The presentation of all methodology and results is very clear.
- A large-scale data collection with a thoroughly curated design of DIVERSE (many participants, many data points, and multi-turn interaction logs).

**Weaknesses:**

- Allowing participants to choose their own topic of conversation can enhance validity of experiment setup (data collection especially), but this is likely to inject some heterogeneity in the task type and difficulty that can affect the pairwise evaluation settings. Although the paper collects LLM-as-judge annotations over several aspects, these variables are not put to the hierarchical TBD model. This can risk that the model assumes all conversations are equally treated, even though some task contexts may inherently require multi-step reasoning of problem solving (than just freely writing creative text). Ignoring this may not reveal quality differences between LLMs in a pairwise setup. If a LLM happens to be matched frequently on tasks that the model is good at, the estimated skill parameter of this LLM may be inflated independent of true capacity it has. The current manuscript does not provide those level of analysis.

- The hierarchical BTD model treats age, ethnicity, and political affiliation as additive effects (scaled by 1/ $\sqrt{3}$). This assumes that each demographic axe contributes independently to preference. However, human preferennce often involves interaction effects. For example, political tone or language can differ between younger and older generations (e.g., younger conservatives vs. older conservatives). The political identity of an individual may differ across racial and ethnic groups. The current manuscript does not consider this interactions of several demographic dimensions in the analysis.

**Questions:**

- Could you provide a breakdown of task types and complexity (from the collected LLM-judge annotations) that each model was exposed to? Do win-rates change substantially when comparing models within the same task type (e.g., reasoning tasks like math, or causal conversation?) Is the overall leaderboard remaining stable when grouping by task complexity or task topic? Some stratification analysis would be highly valuable to strengthen the paper.

- Did you check whether preference patterns differ across combinations of demographics, such as age x politics? A simple heatmap or table showing win/tie rates for age x politics groups can help assess whether those interaction effects are present.

---

> ### Author Response · Authors · 2025-11-18
> **Task Heterogeneity, Demographic Interactions, and LLM Judge Clarification**
>
> Thank you for your review. Below we address the points you raised.
>
> ## Weakness 1 & Question 1: Task Heterogeneity and Task-Type Stratification
>
> You raise an important point about task heterogeneity. This was a deliberate design choice to prioritise ecological validity. Prescriptive prompts risk (1) constraining evaluation to researcher-anticipated use cases, potentially missing capabilities or failures that matter to real users, and (2) requiring participants to engage with unfamiliar topics, compromising judgment reliability. Our goal was to capture natural heterogeneity of real world interactions across the full spectrum of actual use cases.
>
> However, you correctly identify that if certain models were systematically matched with advantageous task types, their skill parameters could be artificially inflated. Our experimental setup used randomised, anonymised models with TrueSkill-based adaptive sampling prioritising high-uncertainty matchups, not specific task types. There was no mechanism for systematic task assignment.
>
> Nonetheless, to address this concern, we conducted a sensitivity analysis examining whether models were disproportionately assigned to specific task categories. We conducted a chi-square test of independence using the top 20 task types to test whether model identity and task type distribution are associated.
>
> The test yielded χ²(494) = 916.89, p < .001, technically rejecting the null hypothesis. However, the effect size is negligible (Cramér's V = 0.035, where 0.1 is considered small). With 40k+ observations, minuscule deviations from perfect uniformity become statistically detectable. The Cramér's V of 0.035 indicates model identity explains <0.1% of variance in task distribution, practically equivalent to random noise, indicating no systematic bias.
>
> **Datasets and leaderboard** (navigate to "Conversation Analysis" > "Data Deep Dive" for task breakdowns):
> - Preference feedback: https://drive.google.com/file/d/1pC8ZQ7AnjZti4ibcXsspSo15OC9Wq2m2/view?usp=sharing
> - Conversation metadata: https://drive.google.com/file/d/1XKwsOtwbM9W4h6Ow58FhwtA7sEHVUIAP/view?usp=sharing
> - Leaderboard: https://diverse-leaderboard-565860601788.europe-west1.run.app/
>
> ## Weakness 2 & Question 2: Interaction Effects in the Hierarchical Model
>
> You raise an important question about whether our additive hierarchical model misses interaction effects between demographic dimensions (e.g., age × politics). Following your suggestion, we conducted a direct empirical analysis of tie rates across all demographic combinations using a two-way ANOVA decomposition. We have added the full interaction analysis to Appendix D with decomposition heatmaps showing observed, expected, and interaction components for all three demographic pairs.
>
> However, to address any potential interactions, our hierarchical model uses partial pooling to learn demographic patterns across combinations. Explicitly modelling all two-way and three-way interactions would require 100+ additional parameters. With median sample sizes of ~400-650 observations per cell for two-way interactions, this is theoretically feasible with hierarchical priors, but the financial and practical feasibility of such an approach is considerably beyond what we are able to achieve in this framework. We therefore chose the simpler additive approximation to prioritise interpretability. We acknowledge this choice means some demographic specific patterns are absorbed into the model's hierarchical structure rather than explicitly parameterised, but we believe the combination of the main effects we specify and the heatmaps added (below) provide a robust characterisation of the data.
>
> ## Clarification: Role of the LLM Judge
>
> To clarify: the LLM judge was never used to generate preference judgments. All pairwise preferences across all five dimensions came directly from human participants. The LLM judge was used exclusively for post-hoc classification of conversation transcripts to generate explanatory metadata on task type, domain, and complexity. This metadata aids understanding but was not an input to the BTD model—rankings are derived exclusively from human judgments.
>
> We excluded these LLM-generated task variables from the primary hierarchical BTD model to (1) keep the model interpretable and focused on demographic effects, our primary research question, and (2) address concerns about reliability and biases of LLM-based classification (Schroeder & Wood-Doughty, 2024; Wataoka, Takahashi & Ri, 2024). We provide them as complementary metadata rather than incorporating them into the primary statistical model.

---

> > ### Comment · Reviewer_n89e · 2025-11-26
> >
> > Thank you for the additional analyses and clarifications that you've made within this short timeframe. I have adjusted my scores. Please reflect your additional analyses on the current manuscript.

---

### Official Review · Reviewer_Kbk1 · 2025-11-07

**Soundness:** 2
**Presentation:** 3
**Contribution:** 3
**Rating:** 6
**Confidence:** 4

**Summary:**

The paper proposes DIVERSE, a large-scale, demographically aware framework for evaluating large language models through human preference data rather than technical benchmarks.

Strengths

- Dataset scale:

The study is based on 106,760 pairwise comparisons from 21,352 participants across 27 language models, which provides substantial empirical depth.

- Methodological rigor:

The hierarchical Bayesian BTD model is statistically sound and appropriate for modeling heterogeneous human preferences.

- Insightful analysis:

The examination of demographic heterogeneity and metric discriminability yields novel and meaningful findings for human-centered LLM evaluation.

Weakness

- Data Availability

The paper briefly mentions data and framework availability in the conclusion but does not provide access at review time. Given the paper’s emphasis on dataset, it would be important to release at least a partial dataset or representative samples during the review process. If the paper is accepted and made public, the authors should clearly commit to releasing the full dataset for research use.


- Representativeness

While the paper makes a valuable contribution, its claims about representativeness and the mitigation of sampling bias are overstated. The participant pool includes only users from the US and UK—two English-speaking, Western countries, failing to capture the broader global, multilingual, and cultural diversity of LLM users. The claim of being “stratified across 22 demographic groups” creates an impression of global inclusiveness, though the scope is in fact regionally constrained. The authors should moderate such claims and clearly state that their findings are representative only within certain contexts. A more cautious framing would enhance the paper.

**Strengths:**

- Dataset scale:

The study is based on 106,760 pairwise comparisons from 21,352 participants across 27 language models, which provides substantial empirical depth.

- Methodological rigor:

The hierarchical Bayesian BTD model is statistically sound and appropriate for modeling heterogeneous human preferences.

- Insightful analysis:

The examination of demographic heterogeneity and metric discriminability yields novel and meaningful findings for human-centered LLM evaluation.

**Weaknesses:**

- Data Availability

The paper briefly mentions data and framework availability in the conclusion but does not provide access at review time. Given the paper’s emphasis on dataset, it would be important to release at least a partial dataset or representative samples during the review process. If the paper is accepted and made public, the authors should clearly commit to releasing the full dataset for research use.


- Representativeness

While the paper makes a valuable contribution, its claims about representativeness and the mitigation of sampling bias are overstated. The participant pool includes only users from the US and UK—two English-speaking, Western countries, failing to capture the broader global, multilingual, and cultural diversity of LLM users. The claim of being “stratified across 22 demographic groups” creates an impression of global inclusiveness, though the scope is in fact regionally constrained. The authors should moderate such claims and clearly state that their findings are representative only within certain contexts. A more cautious framing would enhance the paper.

**Questions:**

1. Dataset release: Can the authors provide example cases or partial data during review to improve transparency? Will the complete dataset be released for research use if the paper is accepted?
2. Demographic scope: Do the authors plan to extend data collection beyond the US and UK to include more diverse populations in future iterations of DIVERSE? Otherwise, it would be better to limit the contexts in the introduction.

---

> ### Author Response · Authors · 2025-11-18
> **Dataset Release and Geographic Scope Clarification**
>
> Thank you for raising these points, they are central to the paper.
>
> ## Dataset Release
>
> The datasets were meant to be included in the submission but it appears you weren't able to access them. We apologise for this oversight. We have now uploaded them to Google Drive for review:
>
> - Preference feedback dataset: https://drive.google.com/file/d/1pC8ZQ7AnjZti4ibcXsspSo15OC9Wq2m2/view?usp=sharing
> - Conversation metadata dataset: https://drive.google.com/file/d/1XKwsOtwbM9W4h6Ow58FhwtA7sEHVUIAP/view?usp=sharing
>
> And we have also made a leaderboard app available at: https://diverse-leaderboard-565860601788.europe-west1.run.app/
>
> At present we have not included the above links in the manuscript due to the need to maintain anonymity during the review process. If accepted we will include links to the HuggingFace hosted datasets in the final version of the paper. So, to directly answer your question, yes, we commit to releasing these datasets publicly after the review process is over. We plan to keep the datasets up to date as we add new models and extend the evaluation to additional languages, demographic groups and countries. We have stated this clearly in Section 6.
>
> ## Representativeness
>
> You are right to call out this limitation, the geographic constraint to the US & UK populations is significant, and we need to be more precise about what we are and aren't claiming.
>
> **What we ARE claiming:**
> We address demographic sampling bias within the US and the UK populations. Platforms like LMArena suffer from self-selection bias where certain demographic groups (e.g. younger, more technical users) are overrepresented. Our stratified sampling approach provides proportional representation across age, ethnicity and political affiliation within these 2 countries, with post-stratification to census data. This is a meaningful improvement over existing human preference evaluations for these specific populations.
>
> **What we are NOT claiming:**
> We do not claim global representativeness. You correctly note that describing "22 demographic groups" can create an impression of global inclusiveness when the scope is actually more limited to 2 English speaking Western countries. This fails to capture the broader linguistic, cultural and geographic diversity of AI users across the world.
>
> Our long term vision for this project is to extend it to other countries, languages and demographic groups and to build towards truly global coverage. The initial US & UK focus was driven by budget constraints and data availability.
>
> In response to your concerns we have now removed and/or qualified any language suggesting global reach and clarify that findings apply to US and UK only, as well as framing the contribution accurately and positioning DIVERSE as demonstrating a methodology for demographically representative evaluation, with an initial implementation in the US & UK. We have also revised the 22 demographic groups framing and make it clear that it pertains to demographic strata within two countries and not global demographics. Examples of this reframing can be found in Sections 2.4 and 5.1. We hope that these revisions address the your concerns about overstatement of representativeness. We appreciate this critique, it will improve the paper's honesty about scope!

---

### Meta-Review · Area_Chair_dPae · 2026-01-05

**Summary:**

Strengths mentioned by the reviewers:
- Dataset is large in scale and valuable for the community.
- Methodological rigor and clear presentation.
- Insightful analysis
- Well motivated and timely.


Weaknesses mentioned by the reviewers:
- The data can not be reviewed as the authors did not provide it to the reviewers. **Addressed.**
- The data may not be representative (global). **Partially addressed.**
- Allowing participants to choose their own topic can make the data heterogeneous. **Partially addressed.**
- Although the paper collects LLM-as-judge annotations over several aspects, these variables are not put to the hierarchical TBD model. This can risk that the model assumes all conversations are equally treated, even though some task contexts may inherently require multi-step reasoning of problem solving (than just freely writing creative text). Ignoring this may not reveal quality differences between LLMs in a pairwise setup. If a LLM happens to be matched frequently on tasks that the model is good at, the estimated skill parameter of this LLM may be inflated independent of true capacity it has. The current manuscript does not provide those level of analysis. **Partially addressed.**
- The hierarchical BTD model is rather simple and does not consider interactions between features. **Addressed**
- Insufficient alignment between the stated problem and the proposed solution. The paper’s introduction highlights two major issues: (1) the dominance of single-metric evaluation and (2) the neglect of subjectivity. However, the proposed solution, evaluating models on five dimensions across demographic strata, resembles running multiple benchmarks, each focusing on a different aspect. While this is a meaningful improvement, it does not fully capture the “subjectivity” aspect claimed in the introduction, since subjectivity extends beyond demographic factors and fixed evaluation axes. **Partially addressed.**
- Brief and unclear data description. For example, it mentions that “each message sent by the participant was delivered to both models simultaneously” and that “a minimum of 3 conversational turns were required,” but it is unclear how the same message could be reused across multiple turns, given the dependency on previous model responses. The paper also states that participants were free to select topics—what kinds of topics did they typically choose? Furthermore, while gpt-4o-mini was used to flag “low-effort inputs,” the definition and examples of such inputs are not provided. Given that the dataset is the paper’s primary contribution, the absence of even a single example conversation or basic statistics (e.g., number of evaluations per stratum, average conversation length) makes it difficult to grasp the dataset’s structure and richness. **Not addressed.** The description is still not sufficient.
- Overgeneralization and overstatement of findings. The paper emphasizes that “age is the most significant demographic factor driving preference heterogeneity,” but this conclusion holds only among the three studied axes (age, ethnicity, politics). The claim should be scoped accordingly. Similarly, while the paper asserts that “a model’s competitive standing can change dramatically depending on the evaluation lens,” Figure 3 shows high correlation across dimensions, implying that rankings are fairly consistent rather than dramatically shifting. These inconsistencies weaken the strength of the argument. **Addressed.**
- Weak overall contribution and presentation quality. The paper feels stretched for a full-length submission. For example, Section 3.4.2 describes an “LLM judge for conversational analysis,” but this analysis is absent from the results. The discussion section repeats much of the results content rather than extending it, which makes it redundant. Some metrics, such as “average rank shifts” (Section 4.2) and “tie percentage” (Section 4.4), are insufficiently defined. These issues collectively reduce the paper’s overall impact and quality. **Partially addressed.**
- Data arise from a study interface rather than organic, in-the-wild use, so prompts and stakes may diverge from real workflows, creating ecological bias; in addition, compared with standard automated benchmarks, this framework is harder to deploy and slower to collect, and the paper could better justify the return on that extra effort. **Partially addressed.**
- The related work section could be broadened to cover more papers on human preference variability and pitfalls in human feedback (e.g., “Human Feedback is Not a Gold Standard,” ICLR 2024; “Towards Understanding Sycophancy in Language Models,” ICLR 2024; “Dissecting Human and LLM Preferences,” ACL 2024). **Addressed.**

Questions raised by the reviewers:
- Dataset release: Can the authors provide example cases or partial data during review to improve transparency? Will the complete dataset be released for research use if the paper is accepted? **Addressed.**
- Demographic scope: Do the authors plan to extend data collection beyond the US and UK to include more diverse populations in future iterations of DIVERSE? Otherwise, it would be better to limit the contexts in the introduction. **Partially Addressed.**
- Could you provide a breakdown of task types and complexity (from the collected LLM-judge annotations) that each model was exposed to? Do win-rates change substantially when comparing models within the same task type (e.g., reasoning tasks like math, or causal conversation?) Is the overall leaderboard remaining stable when grouping by task complexity or task topic? Some stratification analysis would be highly valuable to strengthen the paper. **Partially addressed.**
- Did you check whether preference patterns differ across combinations of demographics, such as age x politics? A simple heatmap or table showing win/tie rates for age x politics groups can help assess whether those interaction effects are present. **Addressed.**
- The compensation rate of £9/hr is described as "recommended" (line 157), yet this falls below the UK minimum wage of £12.21/hr for workers aged 21 and over. Could you clarify how this rate was determined and whether it meets ethical guidelines for research compensation? **Addressed.**
- The paper repeatedly refers to DIVERSE as a "living benchmark" with "continuous evaluation" and claims to "continuously add new models and update rankings." Is there an active platform where ongoing evaluation is happening? If not, these claims seem misleading and should be clarified or removed. **Addressed.**
- The four evaluation dimensions are stated to derive from a pilot study using factor analysis (Section 3.3), but no details about this study are provided. Given these dimensions are central to your analysis, could you describe the pilot study methodology, sample size, and how factor analysis led to these specific dimensions? **Addressed.**

**Reviewer Concerns:**

See above.

**Reviewer Scores:**

- Reviewer Kbk1: $6 \to 6$. Some points where only partially addressed.
- Reviewer N89e: $6 \to 6$. Some points where only partially addressed.
- Reviewer Ba5U: $2 \to 2$. Several important points where only partially addressed or not addressed.
- Reviewer HzhH: $10 \to 10$.
- Reviewer 3vQb: $6 \to 6$.

---

### Decision · Program_Chairs · 2026-01-26

Accept (Poster)